# Modeling Object Dissimilarity for Deep Saliency Prediction

**Bahar Aydemir**[1*]                                                                          *bahar.aydemir@epfl.ch*
**Deblina Bhattacharjee**[1*]                                                          *deblina.bhattacharjee@epfl.ch*
**Tong Zhang**[1]                                                                                  *tong.zhang@epfl.ch*
**Seungryong Kim**[2]                                                                   *seungryong_kim@korea.ac.kr*
**Mathieu Salzmann**[1]                                                                *mathieu.salzmann@epfl.ch*
**Sabine Süsstrunk**[1]                                                                   *sabine.susstrunk@epfl.ch*

[1]*School of Computer and Communication Sciences, EPFL, Switzerland,*     [2]*Department of Computer Science and Engineering, Korea University, South Korea,*     * *The authors have contributed equally.*

**Reviewed on OpenReview:** *https://openreview.net/forum?id=NmTMc3uD1G*

## Abstract

Saliency prediction has made great strides over the past two decades, with current techniques modeling low-level information, such as color, intensity and size contrasts, and high-level ones, such as attention and gaze direction for *entire* objects. Despite this, these methods fail to account for the dissimilarity between objects, which affects human visual attention. In this paper, we introduce a detection-guided saliency prediction network that explicitly models the differences between multiple objects, such as their appearance and size dissimilarities. Our approach allows us to fuse our object dissimilarities with features extracted by any deep saliency prediction network. As evidenced by our experiments, this consistently boosts the accuracy of the baseline networks, enabling us to outperform the state-of-the-art models on three saliency benchmarks, namely SALICON, MIT300 and CAT2000. Our project page is at https://github.com/IVRL/DisSal.

## 1 Introduction

Humans can see only a small portion of their visual field in high resolution. Therefore, we have developed attention mechanisms to identify the most significant parts of a scene. Visual saliency prediction aims to mimic this process via the computational detection of important image regions (Borji & Itti, 2012). It has applications in diverse domains, including image enhancement (Zhao et al., 2014), image quality assessment (Guo et al., 2011), path navigation (Chang et al., 2010) and biomedical imaging (Jiang & Zhao, 2017). Following the seminal work of Itti et al. (1998), a myriad of solutions for visual saliency detection have been proposed, using both handcrafted features (Itti et al., 1998; Cheng et al., 2015) and, more recently, deep neural networks (Vig et al., 2014; Cornia et al., 2016; Huang et al., 2015; Kümmerer et al., 2017; Liu & Han, 2018; Yang et al., 2020). The success of deep learning approaches, typically outperforming handcrafted models, can be attributed to their ability to reason not only about low-level local contrast but also higher-level cues such as the objects present in the scene. In particular, the recent state-of-the-art deep saliency prediction networks (Kümmerer et al., 2017; Liu & Han, 2018; Jia & Bruce, 2020; Linardos et al., 2021) rely on features extracted from networks that were originally trained for object classification. While the importance of higher-level object information for saliency detection is widely acknowledged, reasoning about individual objects, in other words objectness (Chang et al., 2011), is not sufficient. As discussed in (Borji et al., 2013a; Bylinskii et al., 2016; Yildirim et al., 2020), a good saliency estimator needs to model the relative importance of image regions. For example, as illustrated in Figure 1, while an object observed on its own in a scene might be salient, its saliency significantly decreases when surrounded by other objects of the same category Jin et al. (2015). This is evidenced by the physiological study in (MacEvoy & Epstein, 2009) that shows the role of the lateral occipital complex (LOC) in the human visual system. The LOC

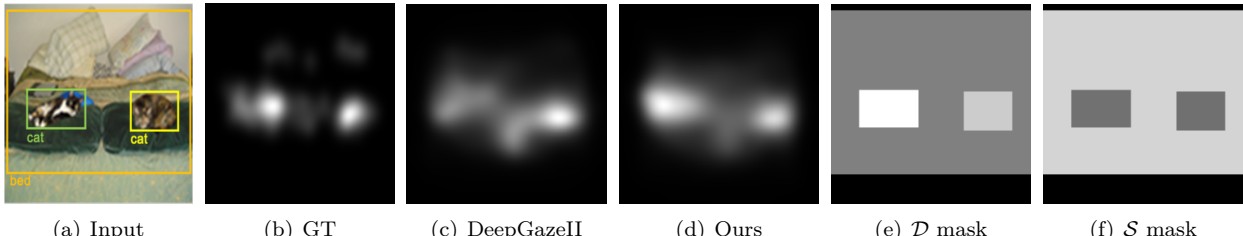

(a) Input      (b) GT      (c) DeepGazeII      (d) Ours      (e) $\mathcal{D}$ mask      (f) $\mathcal{S}$ mask

**Figure 1:** Example of how object **appearance dissimilarity** between multiple objects and **size dissimilarity** affect saliency maps. We show, from left to right, a) the input image from the SALICON benchmark (Jiang et al., 2015), b)the ground-truth fixations, c) the saliency prediction of the baseline DeepGazeII (Kümmerer et al., 2017), d) the saliency prediction of **Our** model that uses both dissimilarity ($\mathcal{D}$) and size masks ($\mathcal{S}$), e) the calculated appearance dissimilarity mask ($\mathcal{D}$) and f) the calculated size mask ($\mathcal{S}$). The cats have similar saliency values in the ground truth whereas in the baseline DeepGazeII the right cat is more salient than the other one. Our model improves the prediction for the left cat with the help of the **appearance dissimilarity** mask, which indicates that the left cat with distinct fur colors is the most dissimilar object. Further, the similar **sizes** of the cats result in similar values in the size mask while the bed is larger. Best viewed on screen and when zoomed in.

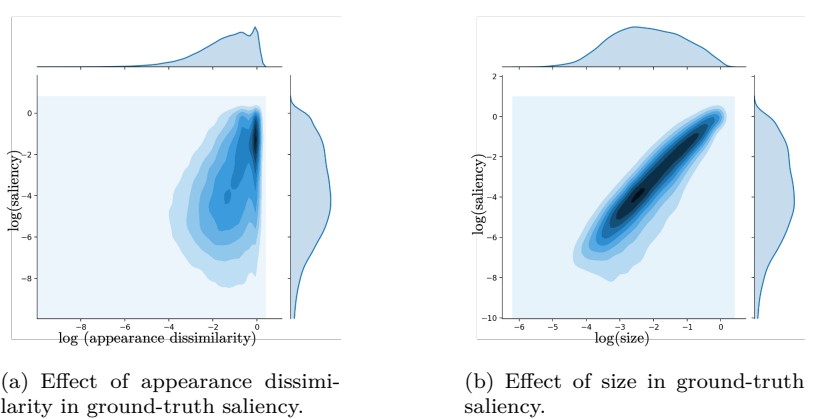

(a) Effect of appearance dissimilarity in ground-truth saliency.

(b) Effect of size in ground-truth saliency.

**Figure 2: Why do we use appearance dissimilarity and size to predict saliency?** To motivate our work, we plot the saliency values against a) appearance dissimilarity and b) size, respectively, on a log scale, showing their presence in the ground-truth saliency maps of the SALICON benchmark. In essence, as the dissimilarities between the objects and their size increases (shown on the x-axis of both the plots), the saliency of the objects increases (shown on the y-axis of both the plots). Here, the concentric rings are the number of object instances. We calculate the appearance dissimilarity and size as shown in Section 3.2. (Best viewed in color.)

To validate the plots statistically, we calculate Spearman's correlation ($r_s$) to determine the relationship between both appearance dissimilarity and size with the saliency values, respectively. There is a strong, positive monotonic correlation between size and saliency ($r_s = 0.817, p < .001$). The correlation between appearance dissimilarity and saliency is weaker but still statistically significant ($r_s = 0.295, p < .001$).

differentiates between multiple objects in a scene by distributing and normalizing the attention across all the objects in the scene. Consequently, this results in a decreased gaze response compared to a scene consisting of a single object. Drawing from this motivation, we introduce a saliency prediction model that reasons not only about multiple objects in a scene but also about the distribution of attention between them.

Furthermore, the human visual system processes visual cues from objects based on their size (Proulx & Green, 2011), such that larger objects in a scene attract more attention. Therefore, in the presence of multiple similar objects, such as instances from the same category, their relative size strongly influences their respective saliency. We leverage this information to additionally model the size of objects in our saliency prediction framework. Note that none of the state-of-the-art deep saliency prediction networks reason about the *contrast* between multiple objects. We model the relative *contrast* between multiple objects in a scene inspired by the presence of object dissimilarity in the ground-truth saliency maps, as motivated by Figure 2. This is what we achieve here via notions of object dissimilarity, by explicitly modeling the appearance and size dissimilarities across the objects observed in the scene.

To this end, we design a detection-guided deep saliency prediction framework that computes a measure of *appearance dissimilarity* between the detected objects together with their relative object sizes, i.e., their *size dissimilarity*. Our architecture then combines these two sources of information with the local features extracted by a convolutional neural network, which rather focus on capturing local contrast. Our approach is general, and thus can be integrated into most state-of-the-art saliency detection networks to improve their accuracy. Note that our approach is inspired by principles observed in the human visual system, including object similarity encoding and attention distribution (normalization).

Our main contributions can be summarized as follows:

- We introduce an object detection-guided model for saliency detection that exploits the contrast between the multiple objects in the scene.

- We show, in particular, that reasoning about the objects' appearance and size dissimilarities boosts the saliency detection accuracy.

- We propose a generic approach; it applies to most modern saliency detection networks and can predict the salient regions in an image by leveraging the predictions from any object detector.

Our experiments on the SALICON (Jiang et al., 2015), MIT1003 (Judd et al., 2009) and CAT2000 (Borji & Itti, 2015) benchmarks demonstrate that our approach consistently improves the results of the baseline saliency networks we build on, for DeepGaze II (Kümmerer et al., 2017), EML Net (Jia & Bruce, 2020) and for UNISAL (Droste et al., 2020). In particular, by using DeepGaze II (Kümmerer et al., 2017) as baseline network, our method allows us to outperform the state of the art on the SALICON benchmark by 4.8% in KLD (Vidyasagar, 2010), 6% in sAUC (Borji et al., 2013b), and 5% in NSS (Peters et al., 2005). We will make our code publicly available.

While we reason about objects, our goal differs from that of salient object detectors (Achanta et al., 2008; Yildirim et al., 2020; Wang et al., 2019) that output binary saliency masks. We do not solely focus on objects but rather aim to produce a continuous saliency map that highlights all the important regions in the input image, which could then be used for different tasks, such as image to image translation (Alami Mejjati et al., 2018), image colorization (Achanta et al., 2008), and image resizing (Achanta & Süsstrunk, 2009).

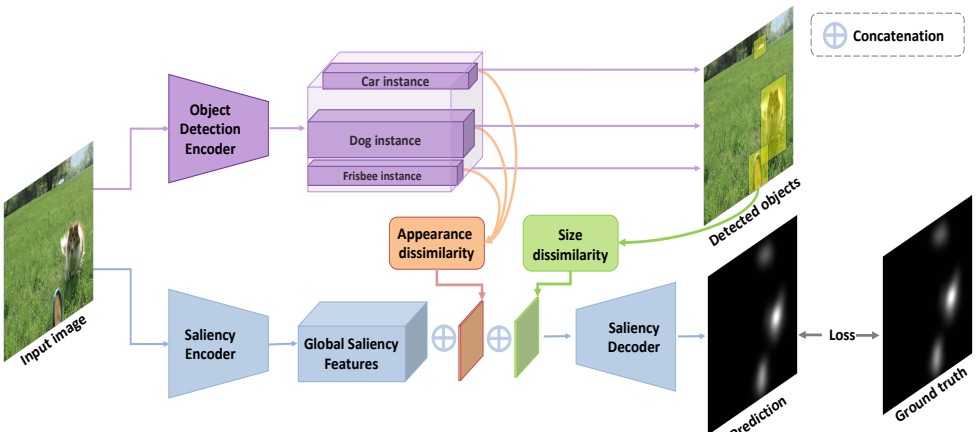

**Figure 3: Overview of the proposed architecture.** We use an object detector to extract object instances. We then pass on these object features to calculate *appearance dissimilarity* (shown in orange), which results in a dissimilarity score for each object instance. The object detection network also outputs a bounding box for each object, which we use to calculate the normalized object *size dissimilarity* (shown in green) for each detection. We then fuse (1) the encoded global saliency features resulting from the saliency encoder, (2) the object appearance dissimilarity features, and (3) the normalized object size dissimilarity features. We train our saliency decoder on this concatenated feature set. We supervise the training with a KLD loss (Vidyasagar, 2010) between the predicted saliency map and the ground-truth one. (Best viewed in color.)

## 2 Related Work

### 2.1 Evolution of Saliency Prediction Algorithms

*What stands out in a scene?* Since the pioneering works of Buswell (1935) and Yarbus (1967), many have attempted to answer this question by predicting the salient image regions that correspond to human visual attention. For example, Itti et al. (1998) proposed a biologically-inspired approach based on the color, intensity, and orientation contrast to simulate human visual attention. Zhang et al. (2008) improved the resulting saliency maps by using self information of visual features; Hou & Zhang (2007) proposed to relate residual features in the spectral domain to the spatial one. Gao et al. (2008) used center-surround contrast of various modalities to classify a region as salient. In contrast to the previous methods that focused on low-level information, Judd et al. (2009) further incorporated mid-level and high-level semantic features, using horizon, face, person, and car detectors. Moreover, Chang et al. (2011), explored the relatedness of objectness and saliency, each of which is known to play a significant part in the study of visual attention. These approaches, however, focus on individual objects, and thus cannot model the contrast between multiple objects.

Furthermore, all the above methods use hand-crafted features, and thus do not benefit from jointly extracting the features, integrating them, and predicting saliency values.

As in most computer vision subfields, this was addressed by training convolutional neural networks (CNNs) for saliency prediction (Vig et al., 2014; Cornia et al., 2016; Huang et al., 2015; Kümmerer et al., 2017; Jia & Bruce, 2020; Cornia et al., 2018; Droste et al., 2020). In particular, Kümmerer et al. (2015) showed that reusing the deep networks trained for object classification significantly improved saliency prediction, thus further evidencing the importance of reasoning about objects, as Judd et al. (2009) already had with non-deep detectors. This trend was then followed by most of the state-of-the-art deep saliency prediction networks, such as SALICON (Huang et al., 2015), Deepgaze II (Kümmerer et al., 2017), DeepFix (Kruthiventi et al., 2017), MLNet (Cornia et al., 2016) and SAM (Cornia et al., 2018), and further extended by Jia & Bruce (2020), whose EML-Net fuses the features extracted from multiple object classification CNNs. Similarly, Linardos et al. (2021) evaluated and combined different object classification backbones for saliency prediction. Nevertheless, these methods only implicitly consider objects via their pre-trained backbones, and, more importantly, do not model the dissimilarities between multiple objects, which strongly affect their respective saliency. Here, we propose to use an object detector to explicitly extract object information, as done by Judd et al. (2009) prior to deep saliency models, but further introduce an explicit reasoning about the differences between the objects in the scene.

### 2.2 Objects and Saliency

The importance of objects in human visual attention has been thoroughly studied from a psychophysical point of view. For example, the work of Russell et al. (2014) was motivated by the studies of Gestalt psychologists (Borji, 2018), arguing that humans perceive objects as a whole before analyzing individual components. Similarly, Nuthman & Henderson (2010) showed that humans tend to look at the center of objects, which typically indicate salient regions because they are easily distinguishable from the background. Einhäuser et al. (2008) showed that object locations predict eye fixations better than low-level features, such as color, image contrast, orientation and motion, although whether this remains true in free viewing conditions has been disputed in (Borji et al., 2013a). Bruce et al. (2016) claimed that objects provide an important guidance to the gaze, which can nonetheless be overwritten by feature contrast. Furthermore, Fan et al. (2018) discussed the relative importance of multiple salient regions, accounting for the influence of emotional objects in visual attention. Also, Zhang et al. (2021) proposed a graph-based saliency prediction model by leveraging object-level semantics and their relationships. Ding et al. (2022) bridged higher-level features to low-level layers with a recursive pathway.

Ultimately, these works confirm that saliency not only arises from low-level information but also from high-level cues, both of which we exploit here.

Specifically, one of the high-level cues we use is relative size. This is motivated by studies that have shown the importance of size in human perception (Wolfe & Horowitz, 2004; Borji et al., 2013c) and that the size

of an object can give information about its geometric and physical constraints, utility and shape (Konkle & Oliva, 2012).Wolfe & Horowitz (2004) classify object size as one of the undoubted guiding attributes of visual attention. Further, Borji et al. (2013c) evidenced this via a psychophysical experiment and showed that object size boosts saliency prediction when linearly combined with a bottom-up saliency predictor. This study further emphasized that neither bottom-up features nor size are sufficient to accurately predict saliency, thus highlighting the importance of combining low-level and high-level cues. We go a step further and incorporate not just size, but also size *dissimilarity* in our model.

The second source of high-level information we use is object appearance dissimilarity. As shown by Todorovic (2010), human perception changes when the context of a visual target is altered without any change to the target itself. That is, a given region in a scene can be either salient or inconspicuous depending on its surroundings. This observation was leveraged by Goferman et al. (2011) by using global and local similarities of image patches to find regions with high contrast, and by Wang et al. (2019) by relying on patch dissimilarity to estimate local and global context. While these bottom-up models were based on handcrafted representations for patch dissimilarity, a few recent works have attempted to incorporate context in deep networks (Liu & Han, 2018; Yang et al., 2020; Kroner et al., 2020). This, however, was achieved via either dilated convolutions (Yang et al., 2020; Kroner et al., 2020) or recurrent units (Liu & Han, 2018), thus essentially focusing on low-level contextual information, without reasoning about object dissimilarities. Although Siris et al. (2021) parallels our work and builds upon our idea of modeling object dissimilarity by taking object semantics in the context of scenes, their method is applied to the task of salient object detection, which is not the task we address in this paper.

Here, we propose to explicitly model these object contrasts via a detection-guided saliency detection strategy. We use the dissimilarity between the network-encoded features to calculate object contrast.

## 3 Methodology

Our goal is to develop a deep saliency prediction network that jointly models low-level information at the global image scale and high-level object-based information, including in particular the dissimilarities between the different object instances observed in the scene. Our method is depicted in Figure 3. We extract global saliency features using a saliency encoder. In parallel, we also identify object instances via an object detection module, from which we compute features explicitly modeling object differences, namely their appearance and size dissimilarities. We then fuse the global features with the dissimilarities-related ones and feed the result to a decoder that outputs a saliency map. We process the input scene in terms of object dissimilarities, spatial layout, which includes objects' size and location, and global context. These processes are inspired by principles of the human visual system. Below, we detail the different components of our approach.

### 3.1 Global Saliency Encoder

Following the state-of-the-art saliency prediction method (Kümmerer et al., 2017), to extract global, image-level saliency features, we use the first 16 convolutional layers of a VGG-19 network (Simonyan & Zisserman, 2015) trained for object classification as an encoder. In addition, we also test our method using the EML Net (Jia & Bruce, 2020) encoder, which comprises the NasNet-Large combined with the DenseNet-160 network pre-trained for object classification. These encode features from the whole image, without accounting for objects' dissimilarities, which is what we focus on below. Note that our approach generalizes to any saliency estimation network based on convolutional feature extractors (Cornia et al., 2016; Yang et al., 2020; Jia & Bruce, 2020; Jiang et al., 2015; Cornia et al., 2018; Kümmerer et al., 2015; Kroner et al., 2020). This part of our network represents the global features in the scene, such as edges, textures, object parts, objects and contextual cues.

### 3.2 Modeling Objects' Contrast

In addition to low-level image contrast, mid-level or high-level cues like object contrast benefit visual attention. To explicitly reason about objects and their dissimilarities, we make use of an object detection module. Specifically, we employ the Single Shot MultiBox Detector (SSD) (Liu et al., 2016), which has the advantage of performing detection in a single stage, using a simple and effective architecture. Note, however, that our

approach generalizes to other detectors, as will be shown in our experiments, where we replace SSD with RetinaNet (Lin et al., 2017).

For each detected object, we crop the features in the last SSD layers using the predicted bounding box $\mathbf{b}_i$. This gives us an instance feature map $\mathbf{x}_i \in \mathbb{R}^{w_i \times h_i \times d}$ for every detected object, with height $h_i$ and width $w_i$, and $d$ channels. Since different objects have different spatial dimensions, we interpolate to bring each such feature map to the same spatial resolution, leading to $\mathbf{f}_i \in \mathbb{R}^{w \times h \times d}$, where $w, h, d$ are the maximum width and height among all detections.

**Object appearance dissimilarity.** Gärdenfors (2004) highlights that a conceptual space can represent perceived information with several feature dimensions. Similarity cues can be obtained by calculating a distance between these representations. The authors interpret perceived dissimilarity as a distance between the objects in a conceptual space, which is central for a large number of cognitive processes. In our work, we use an object detector to map the objects into a conceptual feature space. These extracted object features are then used to calculate the perceived dissimilarity/distance. Furthermore, Mur et al. (2013) highlight that people tend to perceive the dissimilarity of objects based on properties including perceived color, shape, and semantic category. Therefore, drawing from this motivation, we calculate the dissimilarity between the objects by using the cosine distance between the raw object features, which typically include high-level information, such as object semantics and shape (Liu et al., 2016). To model the appearance dissimilarity between every pair of detected objects, we use the normalized cosine distance. Given the sliced feature maps of two object instances, $\mathbf{f}_i$ and $\mathbf{f}_j$, the similarity of these objects is expressed as

$$\text{sim}(\mathbf{f}_i, \mathbf{f}_j) = \sum_{k=1}^{d} \frac{\langle \mathbf{f}_{i_{[:,:,k]}}, \mathbf{f}_{j_{[:,:,k]}} \rangle}{\max\big( \|\mathbf{f}_{i_{[:,:,k]}}\| \cdot \|\mathbf{f}_{j_{[:,:,k]}}\|, \epsilon \big)} \tag{1}$$

where $\mathbf{f}_{i_{[:,:,k]}}$ encodes a feature vector obtained by taking the feature dimension $k$ at every spatial location in $\mathbf{f}_i$, $\langle \cdot, \cdot \rangle$ denotes the inner product of two vectors, and $\epsilon$ is a small constant to avoid division by zero. Directly exploiting pairwise dissimilarities in the network is difficult, as such dissimilarities cannot be arranged in the same topology as the global feature map. To address this, for each object instance, we compute a dissimilarity score

$$\text{diss}_A(\mathbf{f}_i) = \frac{1}{\sum_{j \neq i} \text{sim}(\mathbf{f}_i, \mathbf{f}_j)} \tag{2}$$

We then normalize the dissimilarity scores of the different objects between $[0, 1]$. To fuse the resulting dissimilarity scores with the global features, we replicate the score of each object spatially within its detected bounding box, so as to create a single-channel feature map of the same spatial resolution as the global one. We then fill in the regions not accounted for by any object with zeros, and concatenate the resulting feature map to the global one. In case of overlapping bounding boxes, we take the average. An example dissimilarity map is shown in Figure 4.

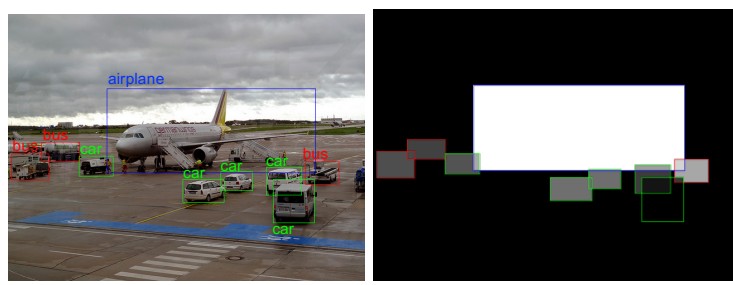

(a) Object Detections      (b) Appearance Dissimilarity Masks

**Figure 4: Visualising object appearance dissimilarity.** For each object, we show its appearance dissimilarity score, calculated based on pairwise feature dissimilarity. The white box corresponding to the single airplane shows the maximum dissimilarity score. When bounding boxes overlap, the average is taken.

**Size dissimilarity.** The size of an object can give additional information about its category, mobility, utilization, shape, geometric and physical constraints (Konkle & Oliva, 2012). Hence, we incorporate this

additional information by providing object size cues to our network. In the human visual system, the neurons in the primary visual cortex can detect changes in visual orientations, spatial frequencies and colors. The location indicated by the highest firing neuron is perceived as a salient location to attract attention, while signals for the other locations are suppressed. Similarly to this normalization, the psychophysical studies of (Carandini & Heeger, 2011) show that local divisive normalization is a key component for humans to learn feature contrast. This normalization prevents the saturation of the multiple signals. It also emphasizes the most prominent signal while suppressing the others. We mimic this mechanism by creating the size dissimilarity masks. To model the size dissimilarity of the detected objects, motivated by normalization in human perception, we normalize their size with a common value. Specifically, we calculate the size dissimilarity of an object as

$$\text{diss}_S(\mathbf{b}_i) = \frac{w_i * h_i}{W * H} \tag{3}$$

where $\mathbf{b}_i$ indicates the bounding box of size $w_i \times h_i$, and $W$ and $H$ are the image width and height, respectively. As with the appearance dissimilarity scores, we replicate the size dissimilarity of each object under the extent of its bounding box, so as to create a one-channel feature map of the same spatial resolution as the global one. We then concatenate it to the global features. This process is illustrated in Figure 5. Note that we calculate size dissimilarity relative to the image size, not relative to the objects. We do not calculate size dissimilarities between the objects as this would assign higher dissimilarity values to small objects in the presence of other larger objects, which differs from the way humans view salient objects in a scene ( Konkle & Oliva (2012)). Specifically, when there are multiple objects from the same category but with different sizes, human attention tends to focus on the objects with larger sizes. This is what we mimic in our model.

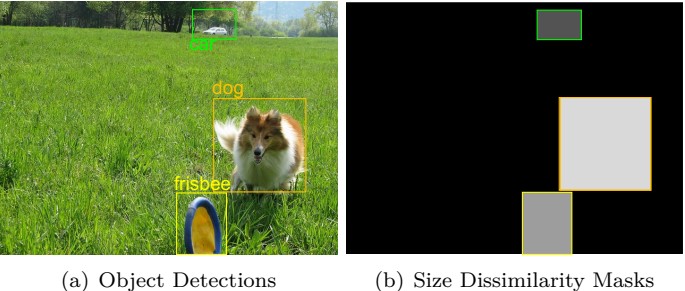

(a) Object Detections      (b) Size Dissimilarity Masks

**Figure 5: Visualising size dissimilarity.** The area of a detected object's bounding box is divided by the image size. This normalized value is associated to the bounding box, with larger bounding boxes having values closer to one, here indicated with a lighter grey color.

### 3.3 Saliency Decoder

Once we have concatenated the appearance and size dissimilarity features discussed above with the global saliency ones, we pass the resulting fused feature map to a saliency decoder. We experiment with the saliency decoder of two state-of-the-art saliency models, namely, DeepGaze II (Kümmerer et al., 2017) and EML Net (Jia & Bruce, 2020). DeepGaze II uses 4 1x1 convolutional layers, also known as the readout layers. We use an upsampling layer to match the channel depth of the concatenated features. Thereafter, we apply a Gaussian kernel followed by a smoothing kernel and a softmax operation to overcome the centre bias and to account for the blurring differences between the ground truth saliency maps across different datasets. In addition, we also test our method using the EML Net decoder, where the feature map is compressed by passing through a single 1x1 convolutional layer followed by a ReLU nonlinearity. The output prediction is then resized to the input size via bilinear upsamling. As with DeepGaze II, these saliency predictions are then passed through a Gaussian kernel and a smoothing kernel. We provide a detailed analysis of our model's performance in conjunction with the above two baselines.

### 3.4 Loss Function

Following common practice (Huang et al., 2015; Kroner et al., 2020; Jetley et al., 2018), we use the Kullback-Leibler divergence (KLD) between the predicted and ground-truth saliency maps as a loss function to train our model. Let $P$ denote the saliency map predicted for one training image and $Q$ the associated ground-truth

map. The KLD is then computed as

$$\text{KLD(P,Q)} = \sum_i Q_i \log \left( \varepsilon + \frac{Q_i}{\varepsilon + P_i} \right) \tag{4}$$

where $i$ iterates over the image pixels and $\varepsilon$ is a small constant to avoid numerical instabilities. Both $P$ and $Q$ are probability distributions, summing to 1. For further quantitative analysis showing the effect of the loss functions in our model, we refer the reader to the supplementary material.

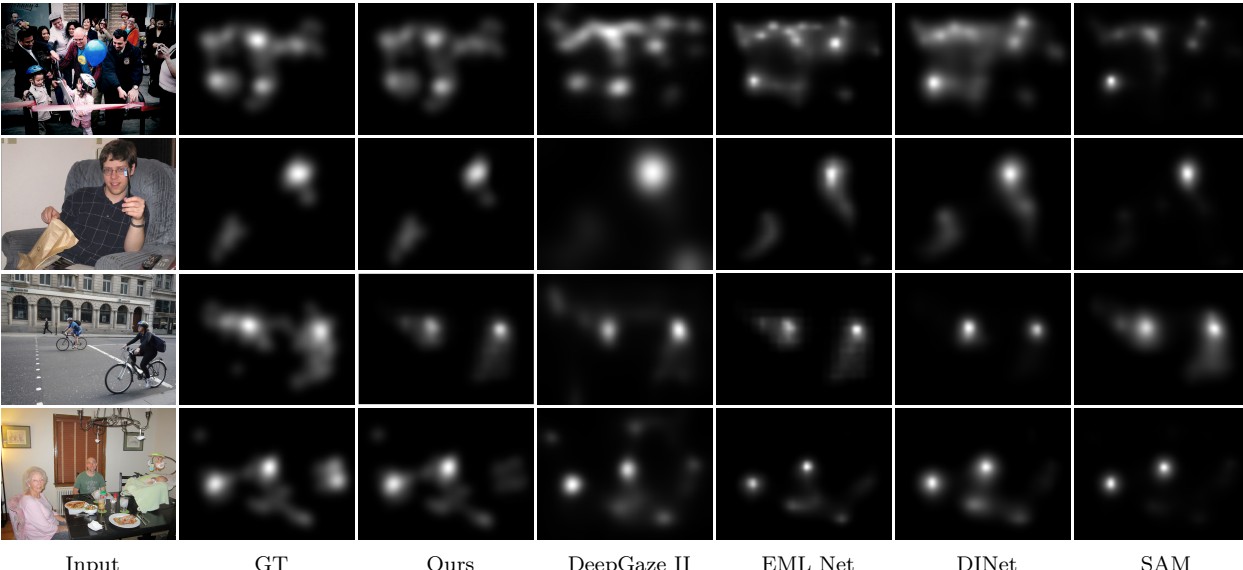

|  Input | GT | Ours | DeepGaze II | EML Net | DINet | SAM |

**Figure 6: Qualitative Results on SALICON (Jiang et al., 2015) validation benchmark.** We show, from left to right, the input image, the corresponding ground truth, saliency maps from our model **(Ours)**, the baseline results from DeepGaze II (Kümmerer et al., 2017), EML Net (Jia & Bruce, 2020), DINet (Yang et al., 2020), and SAM (Cornia et al., 2018), respectively. The top two rows show how object appearance dissimilarity affects saliency. For example, in the top row, the similarity of the objects from the same category (person) decreases their saliency, whereas in the second row, the single occurrence of the person makes him more salient. This appearance dissimilarity also allows our model to predict the saliency of the paper bag, unlike the baseline. We can also capture the saliency of the paper bag that is missed by the baseline. The third row shows the effect of size dissimilarity on saliency. The closest person on the bicycle is the most salient, followed by the second and the third person in decreasing order of their size. The last row shows a typical failure case of our model. It is due to a detection failure (in this case the baby). Note that the baselines also fail on this image.

# 4 Experiments and Results

## 4.1 Experimental Setup

### 4.1.1 Datasets

We report the performance of our methods on three publicly available saliency detection benchmarks. We train our models on 10,000 images of the **SALICON** (Jiang et al., 2015) dataset, which consists of diverse context-rich images from the MS COCO dataset (Lin et al., 2014). Human attention was measured with a crowd-sourced mouse tracking experiment. The resulting pseudo-fixations highly correlate with eye fixations (Jiang et al., 2015). The dataset contains 10,000 training, 5,000 validation, and 5,000 test images, which makes it the largest saliency detection dataset to date. The ground truth of the official SALICON test set is not released but predictions can be submitted for evaluation on the LSUN challenge website[1].

---

[1]https://competitions.codalab.org/competitions/17136

**Table 1: Evaluation results on SALICON (Jiang et al., 2015) and MIT1003 (Judd et al., 2009) validation benchmarks.** We compare state-of-the-art saliency prediction models (Yang et al., 2020; Liu & Han, 2018; Pan et al., 2016; Cornia et al., 2018; Huang et al., 2015; Kroner et al., 2020; Kümmerer et al., 2017; Jia & Bruce, 2020; Droste et al., 2020), respectively. The results in bold and underline show the best and the second best performances. Our method outperforms the state-of-the-art ones on at least four metrics across the two benchmarks.

| Model | SALICON | | | | | | MIT1003 | | | | | |
|---|---|---|---|---|---|---|---|---|---|---|---|---|
| | AUCJ ↑ | KLD ↓ | NSS↑ | CC↑ | sAUC↑ | SIM↑ | AUCJ↑ | KLD↓ | NSS↑ | CC↑ | sAUC↑ | SIM↑ |
| DINet | 0.863 | 0.613 | 1.974 | 0.860 | 0.742 | 0.784 | 0.907 | 0.704 | 2.855 | 0.766 | 0.636 | 0.561 |
| DSCLRCN | 0.869 | 0.637 | 1.979 | 0.831 | 0.736 | 0.715 | 0.880 | 0.725 | 2.813 | 0.750 | 0.624 | 0.530 |
| SalNet | 0.860 | 0.674 | 1.766 | 0.730 | 0.711 | 0.696 | 0.877 | 0.759 | 2.699 | 0.728 | 0.630 | 0.547 |
| SAM | 0.866 | 0.610 | 1.965 | 0.842 | 0.741 | 0.751 | **0.911** | 0.682 | 2.888 | 0.768 | 0.613 | 0.552 |
| SALICON | 0.837 | 0.658 | 1.877 | 0.657 | 0.694 | 0.639 | 0.871 | 0.818 | 2.757 | 0.728 | 0.609 | 0.533 |
| CEDN | 0.875 | 0.583 | 2.011 | 0.829 | 0.724 | 0.777 | 0.895 | 0.660 | 2.525 | 0.790 | 0.630 | 0.592 |
| DeepGaze II | 0.876 | 0.433 | 2.014 | 0.881 | 0.750 | 0.775 | 0.881 | 0.744 | 2.480 | 0.794 | 0.627 | 0.567 |
| EML Net | 0.808 | 0.215 | 2.004 | 0.888 | 0.769 | 0.772 | 0.886 | 0.779 | 2.477 | 0.790 | 0.630 | 0.563 |
| UNISAL | 0.864 | 0.354 | 1.902 | 0.878 | 0.657 | 0.773 | 0.904 | 0.777 | 2.678 | 0.750 | 0.692 | 0.610 |
| **Ours w/ DeepGaze II** | **0.884** | 0.413 | **2.115** | **0.912** | 0.795 | **0.805** | 0.889 | **0.613** | **2.955** | **0.839** | 0.664 | 0.602 |
| **Ours w/ EML Net** | 0.860 | **0.195** | 2.089 | 0.893 | **0.799** | 0.799 | 0.896 | 0.622 | 2.533 | 0.813 | 0.658 | 0.583 |
| **Ours w/ UNISAL** | 0.864 | 0.294 | 1.902 | 0.881 | 0.658 | 0.776 | 0.908 | 0.772 | 2.752 | 0.762 | **0.699** | **0.620** |

We also fine-tune our SALICON-trained models on the **MIT1003** dataset (Judd et al., 2009), which consists of 1003 everyday scenes collected from Flickr and LabelMe, and evaluate them on the commonly used validation partition of MIT1003, and on the official **MIT300** test set, which contains 300 natural images. MIT300[2] is one of the benchmark test sets in the MIT/Tubingen Saliency Benchmark and is commonly used to compare state-of-the-art models. Note that we train, validate and test our models on the same data partitions as all the other state-of-the-art models.

In addition, we fine-tune our model on the **CAT2000** (Borji & Itti, 2015) dataset, which comprises 2000 training and 2000 test images organized in 20 diverse categories, such as *Action, Cartoon, Indoor, Outdoor, Social* and *Line drawings*. As the official test split of CAT2000 is not available anymore, we report the performance of our model and of the state-of-the-art methods on a random split of 25 validation images per category. The same images were used for all methods, and they were not used in the training/validation.

### 4.1.2 Evaluation Metrics

We evaluate saliency predictions according to the following standard metrics used by the community.

**Area Under the Curve (AUC)**: Saliency prediction can be interpreted as classifying fixation vs non-fixation points. The area under the ROC curve shows the trade-off between true positives (TP) and false positives (FP). We use two versions of an AUC metric: **AUCJ** (Bylinskii et al., 2019), which computes the TP and FP rates using all the ground-truth fixation points, and **sAUC** (Borji et al., 2013b), which samples FP points from the ground-truth fixations of other observers as well as from the ground-truth fixations of the same observer over other test images. Therefore, sAUC accounts for inter- in addition to intra-observer variability in the ground-truth fixations to reduce the center bias often present in natural images.

**Normalized Scanpath Saliency (NSS)** (Peters et al., 2005): This metric is computed by comparing the predicted saliency values at the ground-truth fixation points to the average predicted saliency. An NSS score of one indicates that the predicted saliency values at the ground-truth fixation points are one standard deviation above the average.

**Kullback - Leibler Divergence (KLD)** (Vidyasagar, 2010): The KLD encodes the cumulative pixel-wise distance between the predicted and the ground-truth saliency distributions. A KLD score close to zero indicates a better approximation of the ground-truth saliency map by the predicted one.

**Pearson's correlation coefficient (CC)** (Jost et al., 2005): This metric measures the linear relationship between the predicted and ground-truth saliency maps. It ranges from -1 to 1. A CC score close to one indicates a strong linear correlation between the two maps.

**Similarity (SIM) score** (Judd et al., 2012): The similarity score sums, over the pixels, the minimum value between the predicted and the ground-truth saliency maps. Since both of the maps are probability distributions summing to 1, a similarity score of 1 indicates a perfect prediction.

---

[2]https://saliency.tuebingen.ai

### 4.1.3 Implementation

We use a pre-trained object detector and train only the global saliency encoder and decoder. To this end, we use the official SALICON training dataset (Jiang et al., 2015). Our models with the DeepGaze II baseline (Kümmerer et al., 2017) incorporating SSD (Liu et al., 2016), and EML Net (Jia & Bruce, 2020) with SSD (Liu et al., 2016) are validated on the SALICON validation set and is further tested using the official SALICON test set [3]. For MIT1003, we fine-tune and then validate our models using the commonly used validation split in the state-of-the-art models.

For the MIT300 evaluation, we use all of the images from MIT1003 to fine-tune our model. For CAT2000, we use 125 and 50 images across 20 categories to fine-tune and validate our model, respectively. The train-validation split is kept constant across all the models. We refer the reader to the supplementary material for the training details. We implemented our approach using Pytorch and will make our code publicly available. Ultimately, our model takes 234ms on average to process an image, versus 205ms for the baseline global saliency network, Deepgaze II. This relatively small difference, despite our use of an additional detection network, is due to the fact that most of the time is consumed by the VGG sub-network, which is shared by both the saliency encoder and the SSD object detector.

## 4.2 Results

### 4.2.1 Quantitative Results

Table 1 compares the results of our method and of the state-of-the-art baselines on the official SALI-CON (Jiang et al., 2015) validation set and on the MIT1003 (Judd et al., 2009) dataset. On SALICON, our model based on DeepGaze II and SSD outperforms all baselines in terms of AUC-Judd (Bylinskii et al., 2019), NSS (Peters et al., 2005), CC (Jost et al., 2005) and SIM (Judd et al., 2012). Our model with EML Net (Jia & Bruce, 2020) as global saliency network and SSD detector yields the second-best results across most metrics. We further provide a comparison of our model with UNISAL (Droste et al., 2020).

Our model with UNISAL (Droste et al., 2020) as the global saliency network and with the SSD detector (Liu et al., 2016) reports a higher performance across most performance metrics than the vanilla UNISAL, on the SALICON validation set. This shows that our approach is general, effectively strengthening the state-of-the-art saliency prediction networks. Similarly, on the official MIT1003 validation images, our model with DeepGaze II as global saliency network and SSD detector yields the best performance in terms of KLD, NSS and CC. Note that, on this dataset, several methods, such as DINet (Yang et al., 2020) and SAM (Cornia et al., 2018) outperform vanilla DeepGaze II, which suggests that using such networks as global saliency backbone in our approach would allow us to further improve our results. This, however, goes beyond the scope of this paper. Also, note that, while our model with UNISAL incorporating SSD outperforms vanilla UNISAL, it does not perform quite as well as our models with DeepGaze II and EML Net, and thus we decided to exclude it from further comparisons. In Table 2, we further report the results of all models on our self-designed test split of the CAT2000 dataset (Borji & Itti, 2015). As before, our model outperforms the state of the art in most of the metrics. Across the three datasets, our model tends to perform particularly well on distribution-based metrics, such as KLD, SIM and CC. We believe this to be due in part because it was trained using the KLD loss. Furthermore, the fact that our approach consistently outperforms the global saliency network it builds on, i.e., DeepGaze II, EML Net or UNISAL, evidences the benefits of accounting for objects' dissimilarities for saliency prediction.

### 4.2.2 Qualitative Results

In Figure 6, we compare the saliency maps obtained with our method with those of our two main baseline models (Kümmerer et al., 2017; Jia & Bruce, 2020) and of two recent models (Yang et al., 2020; Cornia et al., 2018). Note that our model is able to emphasize object appearance dissimilarity in the presence of objects from the same and different categories. Furthermore, it leverages the size dissimilarity of the objects to improve the predictions. These demonstrate that our network can benefit from object-based information in addition to the local low-level and high-level information extracted by the deep saliency encoder.

---

[3]https://competitions.codalab.org/competitions/17136

**Table 2: Evaluation results on the CAT2000 dataset.** We compare state-of-the-art saliency prediction models DINet (Yang et al., 2020), DSCLRCN (Liu & Han, 2018), SAM (Cornia et al., 2018), SALICON (Huang et al., 2015), CEDN (Kroner et al., 2020), DeepGaze II (Kümmerer et al., 2017) and EML Net (Jia & Bruce, 2020).The results in bold and underline indicate the best and the second best performance, respectively. Our method outperforms the state-of-the-art ones on at least three metrics.

| Model | CAT2000 | | | | | |
|---|---|---|---|---|---|---|
| | AUCJ↑ | KLD↓ | NSS↑ | CC↑ | sAUC↑ | SIM↑ |
| DINet | 0.871 | 0.590 | 2.377 | 0.877 | 0.609 | 0.770 |
| DSCLRN | 0.862 | 0.846 | 2.360 | 0.833 | 0.550 | 0.685 |
| SAM | 0.880 | 0.560 | 2.388 | 0.889 | 0.582 | 0.770 |
| SALICON | 0.861 | 0.866 | 2.340 | 0.803 | 0.529 | 0.648 |
| CEDN | 0.881 | **0.360** | 2.300 | 0.870 | 0.590 | 0.751 |
| DeepGaze II | 0.875 | 0.810 | 1.974 | 0.880 | 0.605 | 0.772 |
| EML Net | 0.874 | 0.971 | 2.380 | 0.880 | 0.591 | 0.752 |
| **Ours w/ DeepGaze II** | **0.888** | 0.519 | 2.207 | 0.893 | **0.626** | **0.782** |
| **Ours w/ EML Net** | 0.883 | 0.669 | **2.398** | **0.895** | 0.608 | 0.764 |

**Table 3: Results on the SALICON test dataset.** We show that modeling objects' appearance and size dissimilarities has a significant influence on saliency, irrespective of the global saliency network used; our method with DeepGaze II outperforms the baseline DeepGaze II (Kümmerer et al., 2017) and our method with EML Net outperforms the baseline EML Net (Jia & Bruce, 2020). The results in bold indicate the best performance relative to the baseline used.

| Model | SALICON Test | | | | | |
|---|---|---|---|---|---|---|
| | AUCJ↑ | KLD↓ | NSS↑ | CC↑ | sAUC↑ | SIM↑ |
| DeepGaze II | 0.867 | 0.931 | 1.271 | 0.479 | 0.787 | 0.742 |
| EML Net | 0.866 | 0.520 | 2.050 | 0.886 | 0.740 | 0.780 |
| **Ours w/ DeepGaze II** | **0.874** | **0.503** | **1.682** | **0.771** | **0.794** | **0.779** |
| **Ours w/ EML Net** | **0.870** | **0.427** | **2.077** | **0.894** | **0.752** | **0.795** |

### 4.2.3 Ablation study

In this section, we evaluate the influence of different components of our approach. Specifically, we study how several ways to encode object information affect performance and compare the use of different global saliency networks with different object detectors. Note that, here, we evaluate the models on the SALICON (Jiang et al., 2015) dataset and we do not consider the centre bias and smoothing post processing operations, as we focus on the contributions of the other components.

We study the effect of centre bias and smoothing in the supplementary material. The results of the ablation study are summarized in Table 5 and discussed below. Note, since the models in Table 5 are not post-processed with centre-bias and smoothing, the results in Table 5 are different than those in Table 1 which includes both centre-bias and smoothing.

**Effect of object features.** Note that, while we argue for the importance of modeling objects' dissimilarities, one could also think of incorporating the individual object's features extracted by the detection network as additional input to the saliency decoder. To evaluate this, we construct a feature block of the same dimensions as the global saliency features, and, for each detected object, place the object features sliced from the detection network to their respective location and fill the rest of this block with zeros.
As shown in Table 5, while using these object features only ($\mathcal{O}$) entails a network with higher capacity, its performance is slightly worse than that of the baseline. This performance is significantly improved by the additional use of our size and appearance dissimilarity representations ($\mathcal{O} + \mathcal{S} + \mathcal{A}$) but nevertheless remains below that of our approach using only the size and appearance dissimilarity ($\mathcal{S} + \mathcal{A}$). This shows that the object themselves do not bring additional information compared to that extracted by the global saliency encoder, unlike size and appearance dissimilarity.

**Effect of size and appearance dissimilarities.** As shown in Table 5, introducing size dissimilarity ($\mathcal{S}$) to the baseline consistently boosts saliency prediction for all metrics. The same can be said of exploiting appearance dissimilarity ($\mathcal{A}$), and, ultimately, the joint benefits of size and appearance dissimilarity ($\mathcal{S} + \mathcal{A}$)

**Table 4: Results on the MIT300 test dataset.** We show that modeling objects' appearance and size dissimilarities has a significant influence on saliency, irrespective of the global saliency network used; our method with DeepGaze II outperforms the baseline DeepGaze II (Kümmerer et al., 2017) and our method with EML Net outperforms the baseline EML Net (Jia & Bruce, 2020). The results in bold indicate the best performance relative to the baseline.

| Model | MIT300 | | | | | |
|---|---|---|---|---|---|---|
| | AUCJ↑ | KLD↓ | NSS↑ | CC↑ | sAUC↑ | SIM↑ |
| DeepGaze II | 0.872 | 0.661 | 2.291 | 0.665 | 0.771 | 0.652 |
| EML Net | 0.876 | 0.844 | 2.488 | 0.789 | 0.746 | 0.675 |
| **Ours w/ DeepGaze II** | **0.881** | **0.468** | **2.351** | **0.784** | **0.780** | **0.667** |
| **Ours w /EML Net** | **0.880** | **0.603** | **2.512** | **0.791** | **0.751** | **0.678** |

**Table 5: Ablation Study to study the effect of the object detector used.** The results in bold and underline indicate the best and the second best performance, respectively. We compare our method under four different settings: A) Our model with DeepGaze II (Kümmerer et al., 2017) as the global saliency network incorparating the SSD (Liu et al., 2016) as the object detector, B) Our model with DeepGaze II as the global saliency network incorporating RetinaNet as the object detector (Lin et al., 2017), C) Our model with EML Net (Jia & Bruce, 2020) incorporating the SSD (Liu et al., 2016) as the object detector and finally, D) Our model with EML Net (Jia & Bruce, 2020) incorporating the RetinaNet (Lin et al., 2017) as the object detector. We see that the main benefits of leveraging objects in saliency prediction come from size and appearance dissimilarities. Based on these results, we exploit size and appearance dissimilarities in the DeepGaze II + SSD variant, which reports the highest performance. Yet, these results show that other, generic object detection network and saliency backbone can be used for our model. Note that the models above include neither Gaussian prior, nor smoothing.

| Model | DeepGaze2 + SSD | | | DeepGaze2 + RetinaNet | | | EML Net + SSD | | | EML Net + RetinaNet | | |
|---|---|---|---|---|---|---|---|---|---|---|---|---|
| | AUCJ↑ | KLD↓ | NSS↑ | AUCJ↑ | KLD↓ | NSS↑ | AUCJ↑ | KLD↓ | NSS↑ | AUCJ↑ | KLD↓ | NSS↑ |
| Baseline | 0.868 | 0.444 | 1.983 | 0.868 | 0.444 | 1.983 | 0.807 | 0.291 | 1.955 | 0.807 | 0.291 | 1.955 |
| Object ($\mathcal{O}$) | 0.863 | 0.443 | 1.975 | 0.864 | 0.445 | 1.978 | 0.801 | 0.292 | 1.912 | 0.802 | 0.293 | 1.920 |
| Size ($\mathcal{S}$) | 0.870 | 0.437 | 2.083 | 0.870 | 0.438 | 2.085 | 0.826 | 0.276 | 1.971 | 0.827 | 0.282 | 1.977 |
| Appearance ($\mathcal{A}$) | 0.871 | 0.436 | 2.089 | 0.870 | 0.437 | 2.087 | 0.829 | 0.266 | 2.010 | 0.829 | 0.269 | 2.003 |
| $\mathcal{O} + \mathcal{S} + \mathcal{A}$ | 0.879 | **0.427** | 2.091 | 0.879 | 0.428 | 2.088 | 0.848 | 0.228 | 2.051 | 0.845 | 0.229 | 2.039 |
| $\mathcal{O} + \mathcal{S}$ | 0.867 | 0.439 | 2.066 | 0.868 | 0.443 | 2.069 | 0.819 | 0.281 | 1.958 | 0.821 | 0.285 | 1.963 |
| $\mathcal{O} + \mathcal{A}$ | 0.870 | 0.436 | 2.082 | 0.870 | 0.437 | 2.080 | 0.829 | 0.270 | 1.994 | 0.830 | 0.273 | 1.980 |
| $\mathcal{S} + \mathcal{A}$ | **0.882** | **0.427** | **2.094** | **0.882** | **0.427** | **2.090** | **0.854** | **0.203** | **2.070** | **0.850** | **0.203** | **2.067** |

yields the best-performing model, as advocated throughout the paper.

**Effect of different object detectors.** As shown in our main experiments, using DeepGaze II as global saliency network yields slightly better results than using EML Net. This remains true if we replace our SSD object detector with a RetinaNet. In fact, as in Table 5, both detectors yield very similar performance.

**Effect of non-detected and mis-detected objects.** As our method relies on detected objects, we study the precision of the objects detected by the detection network. To this end, we explore the robustness of our model against wrongly detected objects and when no objects are detected. We find that our model with DeepGaze II (Kümmerer et al., 2017) as the global saliency network and SSD (Liu et al., 2016) outperforms the vanilla DeepGaze II network (Kümmerer et al., 2017) in the event of mis-detections. Furthermore, our model when tested on images with no objects performs similar to the DeepGaze II global network. This shows that our model remains robust against mis-detections and non-detections, outperforming the baseline in the former case. We refer the reader to the supplementary material for a detailed analysis of the effect of non-detected and mis-detected objects.

## 5 Conclusion

We have presented a saliency detection method that explicitly models the contrast between multiple objects in a content-rich scene. In particular, we have shown that exploiting the appearance and size *dissimilarities* of detected objects in existing saliency detection baselines led to a consistent performance boost on several saliency datasets. In the future, we will consider replacing our object detector with an instance segmentation network, which, despite a higher computational cost, will allow the model to better focus on the objects themselves, excluding background information. This will also allow us to extend our approach to the task of salient object detection.

**Acknowledgement.** This work was supported in part by the Swiss National Science Foundation via the Sinergia grant CRSII5−180359.

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

## A  Modeling Object Dissimilarity for Deep Saliency Prediction (Appendix)

**Overview.**  In this supplementary material, we provide additional qualitative results and ablation studies for a better understanding of the proposed model. The document is structured as follows:

- Section **B**: Implementation Details

- Section **C**: Additional Qualitative Results

- Section **D**: Additional Appearance Dissimilarity Analysis

- Section **E**: Additional Centre Bias and Smoothing Analysis

- Section **F**: Additional Loss Analysis

- Section **G**: Additional Analysis of the Influence of the Detected Objects

- Section **H**: Additional Analysis on Feature Averaging

## B  Implementation Details

### B.1  Extracting Object Features

To explicitly reason about objects and their dissimilarities, we make use of an object detection module. Specifically, we employ the Single Shot MultiBox Detector (SSD) (Liu et al., 2016), which has the advantage of performing detection in a single stage using a simple and effective architecture. Note, however, that our approach generalizes to other detectors, as shown in our experiments in the main paper, where we replace SSD with RetinaNet (Lin et al., 2017).

To account for the different objects' scale, the detections output by SSD typically come from different layers. Therefore, for each detection, we slice the features in the last layer of SSD using the predicted bounding box.  For each prediction with a confidence higher than 0.7, we scale the bounding box coordinates to the last layer of SSD. Then, we slice the corresponding layer with the scaled bounding box coordinates. Note that the absence of fully-connected layers between the source layers and the bounding box predictions allows us to match locations and slice the desired object features.

### B.2  Model Training

As discussed in the main paper, we concatenate the global saliency features with the appearance dissimilarity and relative size features. We then pass the resulting fused feature map to a saliency decoder. Our supervised model training uses the KL Divergence (KLD) (Vidyasagar, 2010) loss. We use two V100, 7 Tflops GPUs with 32 GB memory. The memory and computational cost of training is similar to that of the baseline saliency models we rely on. This is because we do not retrain the object detection network, and the operation in-between the appearance dissimilarity, size dissimilarity, and global feature layers is a simple concatenation. Therefore, our model remains computationally tractable.

During training, we resize all images to 480x640 for the global saliency prediction branch and 300x300 for the object detection one. We do not perform any kind of data augmentation. In the testing phase, we perform the same resizing operations for each image. We initialize our global saliency branch based on DeepGaze II (Kümmerer et al., 2017) and based on EML Net (Jia & Bruce, 2020) with the weights provided by the authors of (Kümmerer et al., 2017) and (Jia & Bruce, 2020), respectively. We use random orthogonal initialization for the decoder layers. Furthermore, we use the Adam optimizer to train the global saliency branch, with an initial learning rate of $10^{-4}$. We set the batch size to 2. We validate the network after each epoch and select the best model from the validation phase to avoid over-fitting. When fine-tuning on MIT1003, we use a batch size of 2 and an initial learning rate of $10^{-5}$.Note that we have tested a large number of hyperparameters using grid search for the batch size and learning rate scheduler. Specifically, the batch size was selected from a space of $[2, 4, 8, 16, 32, 64]$ and the learning rate was selected from a search

space of $[10^{-8}, 10^1]$, with increments of 0.005. Further note that, for the comparison to be fair, the best performing model for each of the baselines was found by the same grid-search procedure.

We also initialize our global saliency branch based on the current state-of-the-art model on the MIT/Tuebingen benchmark, namely UNISAL (Droste et al., 2020), with parameters provided by the authors of (Droste et al., 2020). We use the same parameters and training procedure provided by the authors of UNISAL on the SALICON (Jiang et al., 2015) dataset.

## C   Additional Qualitative Saliency Results

We provide additional qualitative results for our model, DeepGaze II (Kümmerer et al., 2017), EML Net (Jia & Bruce, 2020), DINet (Yang et al., 2020), and SAM (Cornia et al., 2018) on the SALICON (Jiang et al., 2015), MIT1003 (Judd et al., 2009) and CAT2000 (Borji & Itti, 2015) datasets in Figure 10, Figure 11 and Figure 12, respectively. Our model (**Ours**) comprises the DeepGaze II (Kümmerer et al., 2017) backbone, the SSD object detector (Liu et al., 2016), the additional centre bias and smoothing and was trained with the KL Divergence loss (KLD) (Vidyasagar, 2010).

For SALICON (Jiang et al., 2015), the additional results in Figure 10 yet again confirm the benefits of exploiting objects' dissimilarity on saliency. We show results from scenes with multiple objects and from scenes that consist of a single object to demonstrate how dissimilarity affects saliency. The higher performance of our method than the baseline DeepGaze II (Kümmerer et al., 2017), specially, in the event of single objects present in the scene, is due to the size dissimilarity masks of the object and due to the appearance dissimilarity of the object that has both low-level and high-level cues encoded in them. As we have positive values in the size dissimilarity mask of the detected objects, it facilitates the decoder to learn the overall object dissimilarity better, thus improving the saliency. This is also evident from Table 5 in the main paper, where size dissimilarity outperforms the baseline DeepGaze II. Furthermore, the appearance dissimilarity masks has encoded information of not only the high-level cues but also the low-level cues, which in this case, facilitates the decoder to learn a better saliency estimation compared to the baseline DeepGaze II. We see an example of this in Figure 10, last row, where the saliency from the single bird is close to that of the ground truth, whereas the baseline DeepGaze II (Kümmerer et al., 2017), overestimates the saliency of the bird.

Similarly, for MIT1003 (Judd et al., 2009), we show results with either multiple objects or single objects in Figure 11. Lastly, in Figure 12, we show results from 5 different subcategories in the CAT2000 dataset (Borji & Itti, 2015), namely Fractals, Affective, Cartoon, Low Resolution and Noisy. Note that the CAT2000 dataset is very diverse. Therefore, learning the most salient information across different images becomes difficult because the number of image samples belonging to each category is quite small. However, our model learns to predict the saliency for most categories, outperforming the baseline methods.

## D   Additional Appearance Dissimilarity Analysis

The appearance dissimilarity encompasses not only low-level features but also high-level object information. The term appearance dissimilarity has been used in the literature, specifically in image-matching papers, as the distance between features in an image (Hu & Lin, 2016; Kim et al., 2018). Deriving from this definition, we encode appearance dissimilarity as the cosine distance between the raw object features. To further study the effect of the appearance dissimilarity metric on our model, we explore a Singular Vector Canonical Correlation Analysis (SVCCA) metric (Raghu et al., 2017). SVCCA is scale-invariant and captures the semantic proximity of different classes, with similar classes having similar sensitivities. To this end, we study the effect of incorporating SVCCA into our model between the object features obtained from SSD (Liu et al., 2016). Given an object feature pair $(\mathbf{f}_i, \mathbf{f}_j)$ we first perform singular value decomposition (SVD) (Golub & Reinsch, 1970), i.e., $\text{SVD}(\mathbf{f}_i)$ and $\text{SVD}(\mathbf{f}_j)$. This projects the object feature space onto a subspace of $\mathbf{f}_i$ and $\mathbf{f}_j$. We then use Canonical Correlation Analysis (CCA) (Hardoon et al., 2004) between the projected object features. This results in a correlation matrix, from which we extract the correlation values for each object pair in the subspace and fuse them to the global saliency features, along with the size dissimilarity features. We train the model using the KLD (Vidyasagar, 2010) loss as before. The performance of this model is reported in Table 6. Note that for this experiment, the training was done under the same setting

as discussed above on the SALICON (Jiang et al., 2015) validation benchmark. From Table 6, we see that the location based performance metrics, such as AUC-J and NSS, give comparable performance for both SVCCA and cosine distance. However, the distribution based metrics, i.e., KLD and CC (Jost et al., 2005), improve when using SVCCA. This further confirms the generality of our model, as different distance metrics yield comparable performance, consistently outperforming the baseline. As SVCCA facilitates the learning of semantic proximity between object features, we could further train the SSD encoder and the encoder/decoder of our global saliency network using SVCCA, to extract further information about the influence of low-level cues versus high-level ones on saliency. However, this goes beyond the scope of this paper and remains a possible future direction of work.

**Table 6: Ablation Study showing the effect of the different dissimilarity metrics on appearance dissimilarity.** We compare our method used in conjunction with different dissimilarity metrics, namely SVCCA (Raghu et al., 2017) and Cosine distance. We see the effect of different dissimilarity metrics on our saliency predictions. As discussed in the text, the benefits of leveraging objects in saliency prediction come from their size and appearance dissimilarity. Adding size and appearance dissimilarity features outperforms the baseline DeepGaze II (Kümmerer et al., 2017) in both the settings. In particular, we see that CC (Jost et al., 2005) improves for the appearance dissimilarity and hence, for the overall size + appearance model. This is because SVCCA gives the average correlation across aligned directions, thus it is a direct multidimensional analogue of the CC metric. The model trained just on the size features performs similarly to Cosine distance, since it is not affected by the appearance dissimilarity. All the other metrics like AUCJ, KLD and NSS remain comparable, constantly outperforming the baseline. Note that the models above include neither Gaussian prior, nor smoothing. The results in bold and underline show the best and the second best performances, respectively.

| Model | SVCCA (Raghu et al., 2017) | | | | Cosine Distance | | | |
|---|---|---|---|---|---|---|---|---|
| | AUCJ↑ | KLD↓ | NSS↑ | CC↑ | AUCJ↑ | KLD↓ | NSS↑ | CC↑ |
| DeepGaze II(DGII) | 0.868 | 0.444 | 1.983 | 0.881 | 0.868 | 0.444 | 1.983 | 0.881 |
| Ours w/ DGII w/ SSD w/ Appearance | 0.871 | 0.434 | 2.084 | 0.904 | 0.871 | 0.436 | 2.089 | 0.900 |
| Ours w/ DGII w/ SSD w/ Size | 0.870 | 0.437 | 2.083 | 0.897 | 0.870 | 0.437 | 2.083 | 0.897 |
| Ours w/ DGII w/ SSD w/ Appearance w/ Size | **0.882** | **0.426** | **2.091** | **0.909** | **0.882** | **0.427** | **2.094** | **0.905** |

**Relationship between object dissimilarity and saliency.** Here, we qualitatively show the effect of object dissimilarity on saliency, further supporting the ablation study in Table 5 of the paper, where Row $\mathcal{D}$ shows the relationship between object appearance dissimilarity and saliency maps. In Figure 7, we provide a qualitative ablation study by taking masks into account separately to show how the predictions differ depending on the existence of the masks. We evaluate, first, our model trained by using both dissimilarity and size masks (shown in Figure 7(c)); second, a model trained only with dissimilarity masks (shown in Figure 7(d)); and finally, a model trained only with size masks (shown in Figure 7(e)). The results show that the model trained with only dissimilarity (Figure 7(d)) emphasizes the contrast between objects, and that the model trained with only size (Figure 7(e)) yields more spread out attention since the size masks are not very distinct. By contrast, **Our** model that leverages both dissimilarity and size cues learns to combine both of these masks with the features from the global saliency encoder to produce a better prediction (shown in Figure 7(c)).

# E  Additional Centre Bias and Smoothing Analysis

**Effect of centre bias.** Amateur photographers tend to put the object of interest near the center of the image when taking photographs. When looking at a display, we also tend to focus on what is straight ahead of us and rarely look at the peripheral regions of the screen (Tseng et al., 2009). As a result, the ground-truth maps of the different saliency datasets have a strong center bias (see Figure 8). A saliency prediction model that favors this center bias will correctly predict some of the fixations caused by this bias independently of the image content.

Specifically, a model that favors this centre bias will perform better in terms of AUCJ (Bylinskii et al., 2019) metric on a centre biased dataset. This is because AUCJ computes the True Positive (TP) and False Positive (FP) rates using all the ground-truth fixation points. To overcome this, the shuffled AUC (sAUC) (Borji et al., 2013) samples FP fixations from the ground truth of other observers as well as from the ground truth

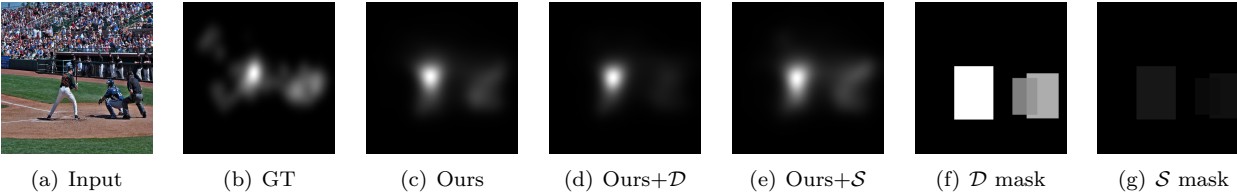

| (a) Input | (b) GT | (c) Ours | (d) Ours+$\mathcal{D}$ | (e) Ours+$\mathcal{S}$ | (f) $\mathcal{D}$ mask | (g) $\mathcal{S}$ mask |

**Figure 7:** Relationship between object dissimilarity and saliency map. We show, from left to right, a) the input image, b)the ground-truth fixations, c) saliency prediction by **Our** model that uses both dissimilarity ($\mathcal{D}$) and size masks ($\mathcal{S}$), d) saliency prediction by Our model that uses only dissimilarity ($\mathcal{D}$), e) saliency prediction by Our model that uses only size ($\mathcal{S}$), f) the calculated dissimilarity mask ($\mathcal{D}$) and g) the calculated size mask ($\mathcal{S}$). Our model that makes use of both the size and dissimilarity between objects yields the best saliency prediction. Best viewed on screen and when zoomed in.

**Table 7: Ablation Study for the effect of centre bias and smoothing.**: We compare our method used in conjunction with different backbone networks, namely DeepGaze II (Kümmerer et al., 2017) and EML Net (Jia & Bruce, 2020) along with the centre bias (CB) and smoothing. We see the effect of centre bias and smoothing on our saliency predictions. We show that sAUC depends on the centre bias of the dataset whereas KLD and CC are affected by smoothing. Bold and underline indicate the best performance and the second best, respectively.

| Model | SALICON | | | | | | MIT1003 | | | | | |
|---|---|---|---|---|---|---|---|---|---|---|---|---|
| | AUCJ↑ | KLD↓ | NSS↑ | CC↑ | sAUC↑ | SIM↑ | AUCJ↑ | KLD↓ | NSS↑ | CC↑ | sAUC↑ | SIM↑ |
| DeepGaze II (DGII) | 0.876 | 0.433 | 2.014 | 0.881 | 0.750 | 0.775 | 0.881 | 0.744 | 2.480 | 0.794 | 0.627 | 0.567 |
| Ours w/ DGII w/o CB w/o Smoothing | 0.882 | 0.427 | 2.094 | 0.905 | 0.767 | 0.793 | 0.887 | 0.675 | 2.891 | 0.806 | 0.631 | 0.579 |
| Ours w/ DGII w/ CB w/o Smoothing | **0.884** | 0.421 | 2.103 | 0.907 | 0.795 | 0.798 | 0.889 | 0.668 | 2.942 | 0.817 | 0.659 | 0.592 |
| Ours w/ DGII w/ CB w/ Smoothing | **0.884** | 0.4130 | **2.115** | **0.912** | 0.795 | **0.805** | 0.889 | **0.613** | **2.955** | **0.839** | **0.664** | **0.602** |
| EML Net | 0.808 | 0.2150 | 2.004 | 0.888 | 0.769 | 0.772 | 0.886 | 0.779 | 2.477 | 0.790 | 0.630 | 0.563 |
| Ours w/ EML w/o CB w/o Smoothing | 0.854 | 0.203 | 2.070 | 0.891 | 0.782 | 0.791 | 0.895 | 0.684 | 2.517 | 0.808 | 0.642 | 0.572 |
| Ours w/ EML w/ CB w/o Smoothing | 0.859 | 0.202 | 2.085 | 0.891 | 0.795 | 0.794 | 0.896 | 0.679 | 2.525 | 0.809 | 0.658 | 0.577 |
| Ours w/ EML w/ CB w/ Smoothing | 0.860 | **0.195** | 2.089 | 0.893 | **0.799** | 0.799 | **0.896** | 0.622 | 2.533 | 0.813 | 0.658 | 0.583 |

of the same observer over other test images. Therefore, sAUC accounts for inter- in addition to intra-observer variability in the ground-truth fixations to reduce the centre bias often present in natural images. The effect of incorporating the centre bias of the datasets into the model demonstrates how sAUC penalizes the centre bias, showing significant improvement in Table 7. Conversely, AUCJ shows no such improvement even after the centre bias is incorporated to the model.

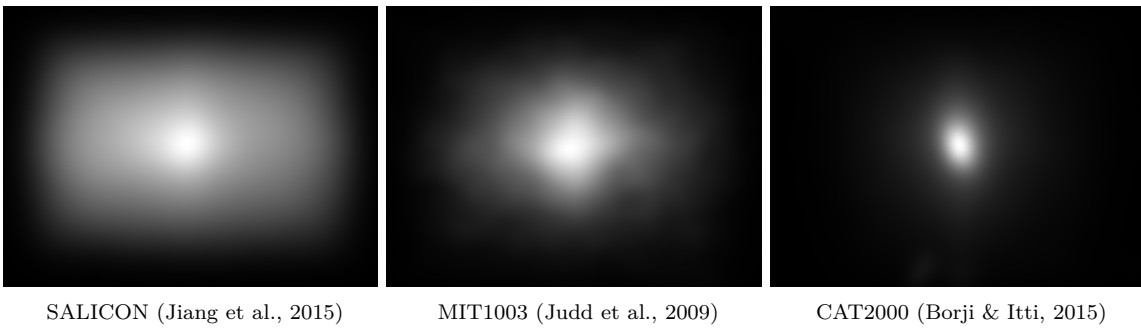

| SALICON (Jiang et al., 2015) | MIT1003 (Judd et al., 2009) | CAT2000 (Borji & Itti, 2015) |

**Figure 8:** Average ground-truth saliency maps for SALICON, MIT1003 and CAT2000 depicting their respective centre biases.

**Effect of smoothing.** Eye tracking experiments record the observers' fixation points on a given image. However, due to the uncertainty of the devices, it is common practice to blur these binary fixation points by a Gaussian kernel corresponding to one degree of visual angle (Le Meur & Baccino, 2012). Thus, the resulting

saliency map has continuous values between 0 and 1. This post-processing step also acts as a regularization and provides robustness to the saliency evaluation as the binary fixation locations from different observers are not likely to overlap. However, the smoothing parameters can have an effect on the models' performance according to the different evaluation metrics. Especially distribution-based metrics, such as KL Divergence (KLD), Pearson's Correlation Coefficient (CC), and Similarity score (SIM), are affected by introducing smoothing to our model, as shown in Table 7.

## F    Additional Loss Analysis

**Effect of different loss functions.**    To study the effect of two different losses, we use the KLD loss (Vidyasagar, 2010), as presented in the main paper, and the EML loss from (Jia & Bruce, 2020). In (Jia & Bruce, 2020), a combination of both the distribution-based metrics and location-based metrics was used to train their saliency prediction model.

Specifically, the EML loss of (Jia & Bruce, 2020) relies on a first component that expresses a modified version of the Pearson's Correlation Coefficient metric (Jost et al., 2005). This component can be written as

$$CC'(P,Q) = 1 - \frac{\sigma(P,Q)}{\sigma(P) \times \sigma(Q)} \tag{1}$$

where $P$ is the predicted saliency map, $Q$ is the ground-truth map, $\sigma(P,Q)$ is the covariance of $P$ and $Q$, and $\sigma(\cdot)$ is the standard deviation. $CC'$ can take values in $[0,2]$.
The EML loss also exploits a modified version of the Normalized Scanpath Saliency (NSS) metric (Peters et al., 2005) expressed as

$$NSS'(P,F) = \frac{1}{N} \sum_i (\bar{R}_i - \bar{P}_i) \times F_i \tag{2}$$

where

$$N = \sum_i F_i \quad (3.1) \qquad \text{and } \bar{P} = \frac{P - \mu(P)}{\sigma(P)} \qquad (3.2) \qquad \text{and } \bar{R} = \frac{F - \mu(F)}{\sigma(F)} \qquad (3.3)$$

and $F$ denotes the ground-truth binary fixation map, $\mu(\cdot)$ and $\sigma(\cdot)$ are the mean and standard deviation, respectively. This term goes to 0 if the predicted saliency map $P$ and the ground truth $Q$ match perfectly.

Finally, the EML loss is defined as the combination of the two above-mentioned terms with the KLD loss, that is,

$$EML_{Loss} = NSS' + CC' + KLD \tag{4}$$

We present the results of our model trained with this EML loss in Table 8. In the same Table 8, we also present the ablation study, yet again, showing the effect of size and appearance dissimilarity when trained with the EML Loss.

## G    Additional Analysis of the Influence of the Detected Objects

As our method relies on detected objects, we study the influence of the precision of the objects detected by the detection network. To this end, we explore the robustness of our model to wrongly detected objects and to not detecting any objects. We do so under three different training settings on SALICON dataset: A) Training our model with ground-truth object bounding box annotations; B) training our model with no detections at all; C) training with predicted detections by the object detector. We then evaluate these models with ground-truth object detections, random detections obtained via the ground-truth annotations of random image from the training set, and no detections at test time.

**Table 8: Ablation Study for models trained with EML Loss described in Section 5.** We compare our method used in conjunction with different backbone networks, namely DeepGaze II  (Kümmerer et al., 2017) and EML Net  (Jia & Bruce, 2020) along with different object detection subnetworks, namely SSD  (Liu et al., 2016) and RetinaNet  (Lin et al., 2017). As discussed in the text, the benefits of leveraging objects in saliency prediction come from their size and appearance dissimilarity. The DeepGaze II + SSD combination performs best, and we refer to that as **ours** approach. Note, however, that for all other combinations, adding size and appearance dissimilarity features outperforms the respective baselines (for fairer comparison, none of the models include Gaussian prior or smoothing). Bold and underline indicate the best performance and the second best one, respectively.

| Model | DGII + SSD | | | DGII + RetinaNet | | | EML Net + SSD | | | EML Net + RetinaNet | | |
|---|---|---|---|---|---|---|---|---|---|---|---|---|
| | AUCJ↑ | KLD↓ | NSS↑ | AUCJ↑ | KLD↓ | NSS↑ | AUCJ↑ | KLD↓ | NSS↑ | AUCJ↑ | KLD↓ | NSS↑ |
| Baseline | 0.866 | 0.440 | 1.989 | 0.866 | 0.440 | 1.989 | 0.808 | 0.222 | 2.042 | 0.808 | 0.222 | 2.042 |
| Object ($\mathcal{O}$) | 0.862 | 0.446 | 1.982 | 0.863 | 0.444 | 1.985 | 0.802 | 0.228 | 1.977 | 0.803 | 0.230 | 1.983 |
| Size ($\mathcal{S}$) | 0.870 | 0.434 | 2.085 | 0.870 | 0.435 | 2.087 | 0.827 | 0.218 | 2.057 | 0.827 | 0.220 | 2.061 |
| Dissimilarity ($\mathcal{A}$) | 0.871 | 0.430 | 2.092 | 0.870 | 0.436 | 2.088 | 0.829 | 0.215 | 2.066 | 0.830 | 0.217 | 2.064 |
| $\mathcal{O} + \mathcal{S} + \mathcal{A}$ | 0.879 | 0.424 | 2.094 | 0.879 | 0.425 | 2.090 | 0.848 | 0.205 | 2.079 | 0.845 | 0.208 | 2.071 |
| $\mathcal{O} + \mathcal{S}$ | 0.867 | 0.436 | 2.069 | 0.868 | 0.439 | 2.073 | 0.819 | 0.220 | 2.045 | 0.821 | 0.221 | 2.049 |
| $\mathcal{O} + \mathcal{A}$ | 0.870 | 0.432 | 2.085 | 0.870 | 0.436 | 2.081 | 0.828 | 0.218 | 2.055 | 0.828 | 0.220 | 2.052 |
| $\mathcal{S} + \mathcal{A}$ | **0.882** | **0.422** | **2.097** | **0.882** | **0.423** | **2.093** | **0.852** | **0.200** | **2.101** | **0.849** | **0.201** | **2.097** |

The results of this study are shown in Table 9. Note that the model that is trained on the predicted objects and tested on the predicted detections learns how to make use of the objects' appearance and size dissimilarities. This is the model we use throughout the paper. The model trained with the ground-truth bounding box annotations and tested on the ground-truth objects provides an upper bound for the performance of our model, giving the optimal training and testing setting. In the event that our model doesn't detect any object, it falls back to the predictions from the baseline model, as evidenced by comparing the results of the model trained with predicted objects and evaluated on no detections with those of the model trained and tested without detections. Furthermore, in the presence of misdetections, our model still outperforms the baseline DeepGaze II, as evidenced by the results of the models trained on predicted objects and tested on random objects. It was seen that, the model learns a robustness against the random objects which can be assumed as noisy detections. In order to validate the robustness of the method, we tested the model with many different False Positives and False Negatives, and it was seen to report a robust performance. This learnt robustness also influences the model, trained/tested on predicted/ predicted objects, respectively. The reason behind this robustness against misdetections is that the SSD predicts some False Positives and False Negatives during training and hence, our model learns a robustness against random objects when it is evaluated on such random objects. However, we see that the model trained/tested on Ground Truth/ random objects, respectively, is not robust against misdetections. This is because we train the model without any False Positives or False Negatives (noise) and evaluate them on random objects at test time. This introduces a lot of False Positives and False Negatives, thus reporting a lower performance. This allows us to conclude that our model remains robust against misdetections and non-detections, outperforming the baseline in the former case.

## H   Additional Analysis on Feature Averaging

We show the performances of our method under the following two experimental settings in Table 10: 1) No averaging within the detected boxes, after computing the pixel-wise cosine dissimilarity; and 2) Reordering the mean operations to leverage the mean features to compute dissimilarity, where the spatial average within the bounding box is taken and then the cosine dissimilarity across the features is calculated. In essence, we do not compute the mean operation in setup (1) and reorder the mean operations in setup (2).

**Qualitatively**, we show in Figure 9 that our dissimilarity mask preserves the contrast cues better than reordering or not computing the mean operations. This leads to improved saliency predictions. Moreover, fine-grained dissimilarity calculation provides different values per pixel in the masks, which results in less robust maps.
In particular, we represent each pixel with a feature vector of size $w \times h \times d$. To compare these feature maps channel-wise, we calculate the cosine similarity along the depth dimension $d$ for each pixel. This results in a

**Table 9: Ablation Study to test the effect of non-detected and mis-detected objects.** Bold and underline indicate the best performance and the second best one, respectively. The results in italics indicate the baseline DeepGaze II (Kümmerer et al., 2017). We train our model under three different training settings: A) with ground truth (GT) object bounding box annotations, B) with no detections at all, C) with predicted detections from the object detector. We evaluate them on GT, random detections obtained via the ground truth annotations of a random image from the training dataset, and no detections at test time. Note that the model that is trained on the predicted objects and tested on the predicted detections learns how to make use of the appearance and size dissimilarities, and is our model of choice. When there is no detection, this model performs similar to the No Detections case shown in italics, which is our baseline. Note that the models above include neither Gaussian prior, nor smoothing.

| Train | Test | AUCJ↑ | KLD↓ | NSS↑ | CC↑ | SIM↑ |
|---|---|---|---|---|---|---|
| GT | GT | **0.882** | **0.425** | **2.094** | **0.909** | **0.795** |
| | Random | 0.866 | 0.450 | 1.980 | 0.830 | 0.768 |
| | No Detections | 0.868 | 0.444 | 1.983 | 0.833 | 0.769 |
| | Predicted | 0.882 | 0.429 | 2.094 | 0.895 | 0.791 |
| No Detections | No Detections | *0.868* | *0.444* | *1.983* | *0.833* | *0.769* |
| | Predicted | 0.866 | 0.449 | 1.980 | 0.832 | 0.768 |
| Predicted | Random | 0.881 | 0.433 | 2.091 | 0.888 | 0.790 |
| Objects | No Detections | 0.868 | 0.444 | 1.983 | 0.833 | 0.769 |
| | Predicted | 0.882 | 0.427 | 2.094 | 0.905 | 0.793 |

**Table 10: Effect of averaging of features on the SALICON benchmark.**

| Model | SALICON validation set | | | |
|---|---|---|---|---|
| | KLD↓ | NSS↑ | CC↑ | SIM↑ |
| DeepGaze II | 0.433 | 2.014 | 0.881 | 0.775 |
| **Ours w/o averaging: Setup 1** | 0.417 | 2.114 | 0.910 | 0.803 |
| **Ours w/ reordered averaging: Setup 2** | 0.416 | 2.114 | 0.911 | 0.804 |
| **Ours w/ DeepGazeII: paper version** | **0.413** | **2.115** | **0.912** | **0.805** |

vector with a different value for each pixel, which yields a mask as per the experimental setup (1), shown in the Figure 9(b). Note that the masks are noisier without averaging. For the mask in the Figure 9(c), we take the spatial average along $(w, h)$ to create one feature vector for each bounding box with size $(1, 1, d)$. Then, we compute the cosine similarity along the channels. For the remaining experimental setting, we obtain **Our** dissimilarity mask, shown in the Figure 9(d), by taking the average within each bounding box in the spatial $(w, h)$ dimensions.

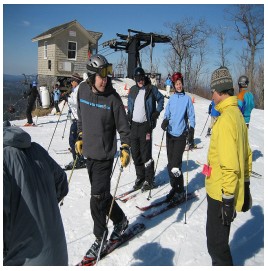 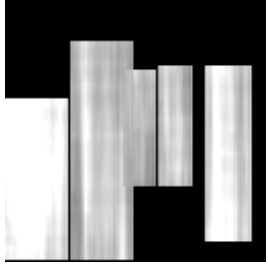 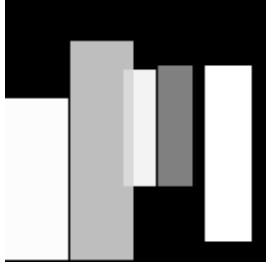 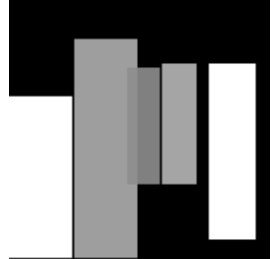

(a) Input  (b) Without average  (c) w/ Reordered averages  (d) Our dissimilarity mask

**Figure 9:** Effect of averaging of features. We show, from left to right, a) the input image, b) the saliency prediction of the model that does not compute the average after calculating the cosine dissimilarity, as described in setup (1), c) the saliency prediction by the reordered mean operation as described in setup (2) and finally, d) the saliency prediction by **Our** model that uses the dissimilarity score between the features before spatially averaging them, respectively. **Our** model outperforms the other averaging techniques. Best viewed on screen and when zoomed in.

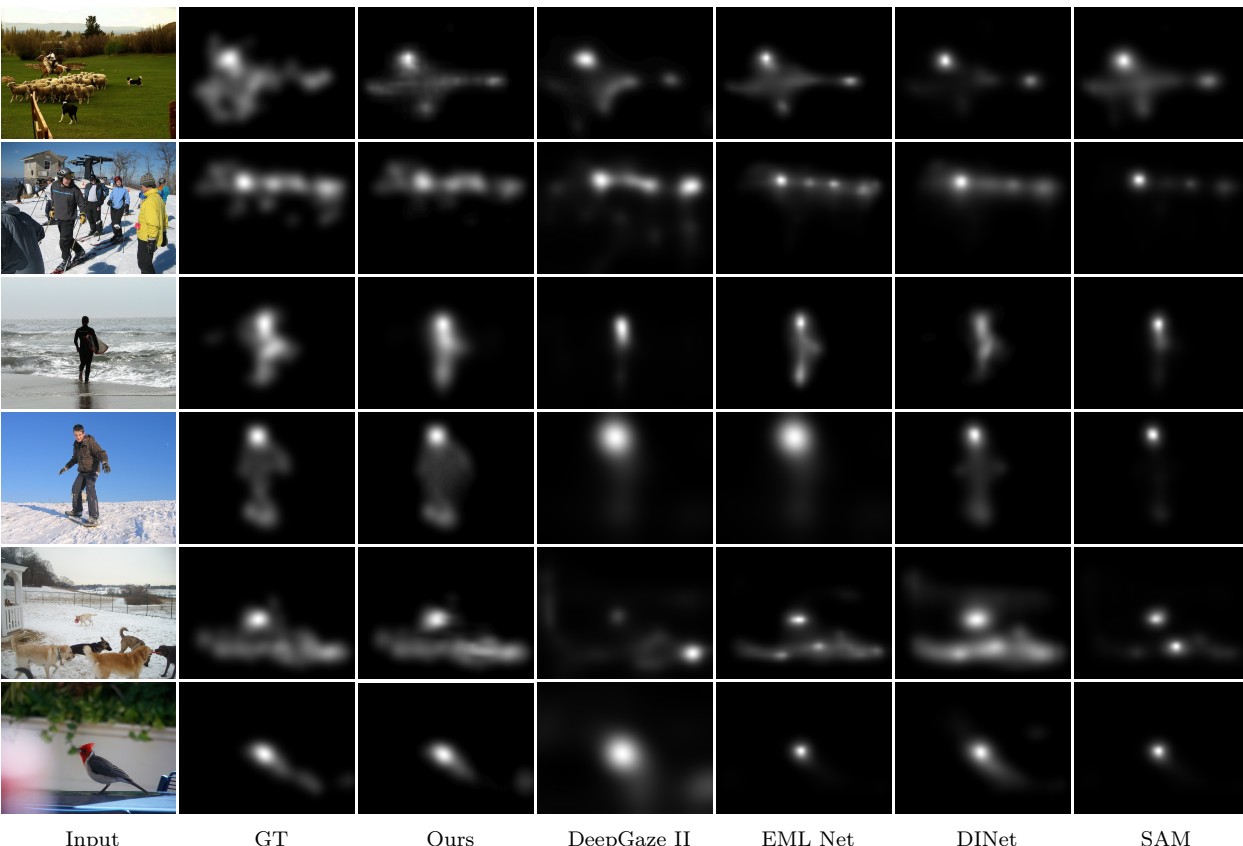

| Input | GT | Ours | DeepGaze II | EML Net | DINet | SAM |

**Figure 10: Additional Qualitative Results on SALICON  (Jiang et al., 2015).** We show, from left to right, the input image, the corresponding ground truth, saliency maps from our model **(Ours)**, the baseline results from DeepGazeII  (Kümmerer et al., 2017), EML Net (Jia & Bruce, 2020), DINet (Yang et al., 2020), and SAM (Cornia et al., 2018), respectively. The results show how objects' dissimilarity affects saliency. The top two rows show how object appearance dissimilarity affects saliency. For example, in the top row, the similarity of the objects from the same category (sheep) decreases their saliency, whereas the single occurrence of the person makes him more salient. Similarly, in the second row, the similarity of the objects from the same category (person) decreases their saliency. Whereas in the third row and the fourth row, the dissimilarity of the person with the surf-board and the skate-board makes him more salient, respectively. Note that the detection of the objects in our model facilitates this whereas the baseline DeepGazeII fails to do so. Similarly, in the fifth row, the similarity of the objects from the same category in the foreground (dogs) decreases their saliency compared to the single dog in the middle, whereas in the last row, the single bird is highly salient. Note that the baseline DeepGazeII model overestimates the saliency in the last row, whereas our model detects the bird and estimates it's saliency close to the ground truth. (Best viewed on screen).

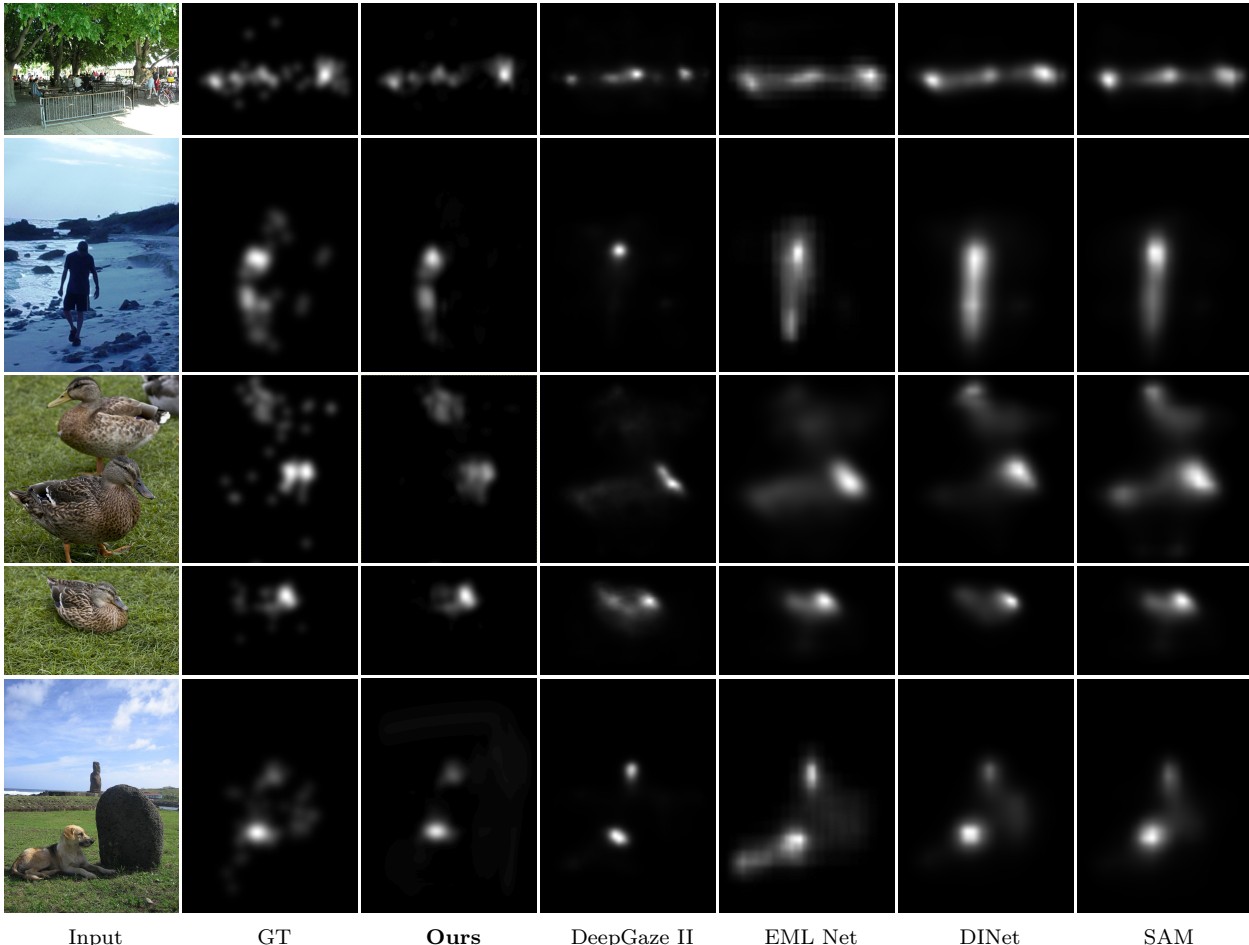

| Input | GT | **Ours** | DeepGaze II | EML Net | DINet | SAM |

**Figure 11: Qualitative Results on MIT1003  (Judd et al., 2009).** We show, from left to right, the input image, the corresponding ground truth, saliency maps from our model **(Ours)**, the baseline results from DeepGazeII (Kümmerer et al., 2017), EML Net (Jia & Bruce, 2020), DINet (Yang et al., 2020), and SAM (Cornia et al., 2018), respectively. The first row shows how both appearance and size dissimilarity affect saliency. For example, the similarity of the objects from the same category (person) decreases their saliency in the left and centre of the image, whereas in the right of the same image the man is more salient than the woman because of his size. In the second row, the single person is highly salient compared to the rocks that are not. Similarly, the similarity of the objects from the same category (duck) in the third row decreases their saliency, whereas in the fourth row, the single duck is more salient. The last row shows a typical failure case of our model. It is due to a detection failure (in this case the rock). Note that the baselines also fail on this image. (Best viewed on screen).

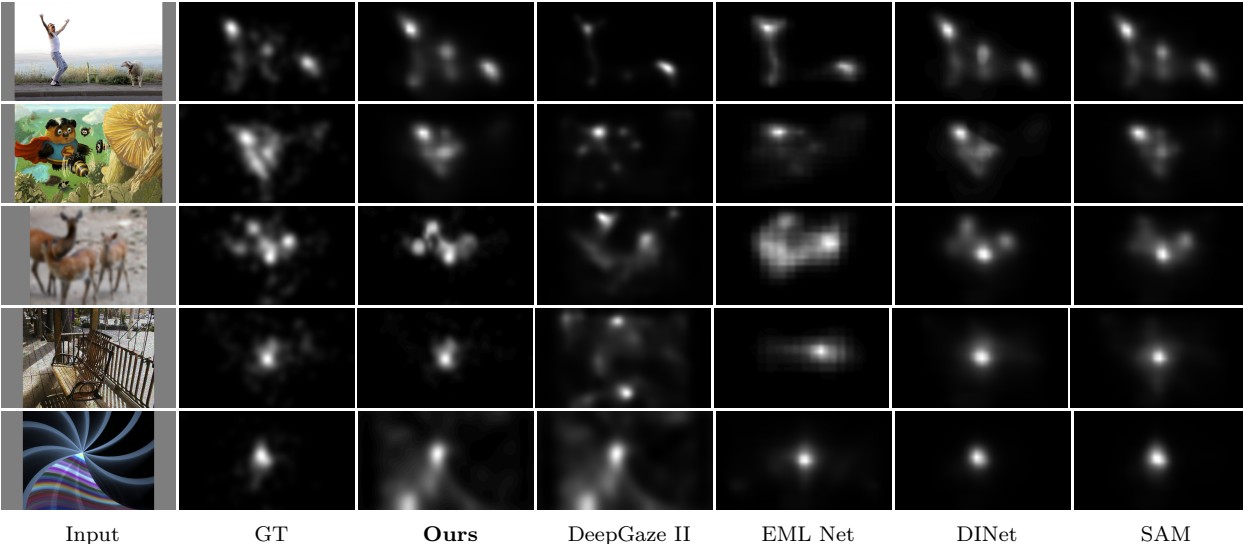

|  | Input | GT | **Ours** | DeepGaze II | EML Net | DINet | SAM |

**Figure 12: Qualitative Results on CAT2000 (Borji & Itti, 2015).** We show, from left to right, the input image, the corresponding ground truth, saliency maps from our model **(Ours)**, the baseline results from DeepGazeII (Kümmerer et al., 2017), EML Net (Jia & Bruce, 2020), DINet (Yang et al., 2020), and SAM (Cornia et al., 2018), respectively. The first row shows the Affective category, the second row shows the Cartoon category, the third row shows the Low Resolution category, the fourth row shows the Noise category and the last row shows the Fractal category, from CAT2000 respectively. As seen from the results, our model performs quite close to the ground truth and outperforms the other state-of-the-art baselines. The first and second row both show the effect of dissimilarity and size on saliency. The third row shows the effect of dissimilarity on saliency when there are multiple objects from the same category (deer) present in the scene. The fourth row is an example from the Noisy category. In the last row, there is no object present in the scene and our object detector fails on such images, and thus our performance is similar as the baseline DeepGaze II. (Best viewed on screen).

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
