# OpenReview forum: "Modeling Object Dissimilarity for Deep Saliency Prediction"
_TMLR — Accepted by TMLR_

### Review · Reviewer_Mvt1 · 2022-08-11

**Summary Of Contributions:**

The paper introduces a more accurate algorithm to predict the human eye-fixation saliency map by leveraging the appearance and size similarity among objects. Results show improvements in saliency prediction datasets with natural images and free-viewing human subjects.

**Broader Impact Concerns:**

No broader impact concerns as far as I know.

**Requested Changes:**

I would encourage the authors to address all the aforementioned weaknesses.

**Strengths And Weaknesses:**

The paper makes a compelling point that taking into account the similarity across objects in the image is quite relevant for predicting human eye-fixations. The ideas are clearly presented and easy to follow. Also, the reported results are quite convincing. I have enjoyed reading the paper.

I would like to highlight the following aspects that in my opinion should be taking into account more meticulously than in the current version:

- The motivation for analyzing the similarity across objects is brought up by citing several papers in neuroscience. However, nothing much is explained about these previous works in the paper. If the reader is not familiar with these previous works, then the reader misses the core evidence that motivates the paper. I would suggest the authors to motivate the study in a more self-contained manner, such that the reader does not need to be familiar with neuroscience literature.

- Also related to the motivation of the paper: most papers in neuroscience need to be taken into account carefully, as the evidence they present is highly dependent on the stimuli and other experimental setup choices. It is common that small changes in experimental details lead to different conclusions. Thus, the paper would do a better job convincing the reader about the importance of taking into account the similarity across objects, if the paper would show evidence from the tested datasets. This could be done by taking statistics that depict more clearly the relationship between object similarity and saliency maps. In the current version of the paper, it is shown that a deep neural network model is capable of capturing such relationships, but it is unclear how prominent these are and what are their characteristics (eg. does the model fully capture the relationship between object similarity and saliency map?).

- The qualitative results do not clearly show how taking into account the similarity of the objects leads to better accuracy. I think the paper would benefit from a figure that relate the qualitative results and the motivation of the paper more clearly.

- The paper does not comment on the generalization capabilities of the model when tested in synthetic stimuli (eg. the stimuli used in some of the cited neuroscience papers, such as images that contain different colored shapes), or when the human subjects are not free-viewing the image.

- I think the number of hyperparameters tested is very narrow and it unclear that the presented results may be dependent on them. Note that it could be that when adding the similarity across objects in the model, the hyper-parameter choice is better. I think it is necessary to check a larger number of hyperparameters to make sure that the results reported are not dependent on them.

---

> ### Author Response · Authors · 2022-08-29
> **Response to Reviewer Mvt1**
>
> Thank you for your time to review our work and for all your valuable feedback. We have incorporated
> your different comments in our updated version, and re-worded/corrected some unclear sentences. For
> better readability of our revision, we have *highlighted in blue* the main modifications we have made.
> Below, we respond to each of your comments.
> 1. **Elaborating neuroscience study.** We have discussed the physiological studies to show the
> appearance and size dissimilarity motivations in a self-contained manner in our revised version.
> We have further elaborated on how normalization across objects is embedded in our attention
> system. We defer the reviewer to Section 3.2 (pages 5- 6) in the revised paper.
> 2. **Relationship between object dissimilarity and saliency map.** We show the statistical
> metrics of AUC, KLD and NSS in Table 5 on page 12 of the paper, where row $\mathcal(D)$ shows the
> relationship between object appearance dissimilarity and saliency maps. We provide additional
> analysis in Section D of the appendix of our revised paper (pages 19-20). Specifically, in Figure
> 6 on page 19 of our revised paper, we provide an ablation study by taking masks into account
> separately to show how the predictions differ depending on the existence of the masks. We
> evaluate, first, our model trained by using both dissimilarity and size masks; second, a model
> trained only with dissimilarity masks; and finally, a model trained only with size masks. Our
> results show that the model trained with only dissimilarity emphasizes the contrast between
> objects, and that the model trained with only size yields more spread out attention since the size
> masks are not very distinct. By contrast, Our model that leverages both dissimilarity and size
> cues learns to combine both of these masks with the features from the global saliency encoder to
> produce a better prediction.
> 3. **Qualitative results for similarity of objects.** In Figure 7, Section D of the appendix of
> our revised paper (page 20), we show the saliency predictions of the baseline DeepGazeII model
> and of Our model with the dissimilarity $\mathcal(D)$ and size masks $\mathcal(S)$. We see that leveraging both
> dissimilarity and size, as done by Our model, improves the saliency prediction. We defer the
> reviewer to Figure 9 of the appendix for further details.
> 4. **Generalizability of the model on synthetic stimuli.** For the synthetic stimuli, we report
> the results in Table 2 on page 10 of the paper, as the CAT2000 benchmark includes a "Pattern"
> category comprising synthetic stimuli. From Table 2, page 10 of the paper, we see that Our model
> consistently outperforms the baseline DeepGazeII model. Furthermore, in the supplementary zip
> file that we have uploaded, we provide additional qualitative analysis on synthetic stimuli for both
> Our model and DeepGazeII when they are fine-tuned to the synthetic stimuli. The results show
> the generalizability of Our model on such stimuli, as we consistently outperform the baseline
> DeepGazeII. We defer the reviewer to the supplementary zip file for further details.
> Nonetheless, **other tasks than free-viewing**, such as object search or memorability assessment,
> can create a heavy bias towards the objects. These tasks can affect the attention shifts by creating
> dependencies between subsequent images. Therefore, we only focus on the free-viewing task for
> image saliency prediction, similarly to the baseline papers.
> 5. **Hyperparameters search.** We have tested a large number of hyperparameters using grid
> search for the batch size and learning rate scheduler. Specifically, as mentioned in Section B
> of the appendix of the paper (page 17), we use a learning rate of $10^{−4}$, which was selected by
> grid search from a search space of $[10^{−8}, 10^{1}]$. Note that, for the comparison to be fair, the
> best performing model for each of the baselines was found by the same grid-search procedure.
> Therefore, we believe that we have provided a fair comparison of the models regardless of the
> hyperparameter choices.
>
> We hope that we have successfully addressed all of your comments. We are looking forward to hearing
> from you.

---

> > ### Comment · Reviewer_Mvt1 · 2022-09-17
> > **Response**
> >
> > Thanks for provided detailed responses. I add some clarification and comments here:
> >
> > 1. Ok!
> > 2. I think my initial comment was not clear. My suggestion was to relate the ground-truth saliency map with the object dissimilarity, ie. showing that the relationship exists in the data and hence it is worth investigating how to leverage this relationship. This was just a suggestion to motivate the paper. It is up to the authors to do it or not. In my opinion will make the paper much more convincing as it will show that since the relationship exist in the ground-truth data, adding mechanisms in the model to capture it would necessarily lead to improvements.
> > 3. Thanks for the clarification. This figure seems to me a much better opener to the paper than current Figure 1 because it actually depicts the key idea of the paper. Current Figure 1 just depicts that the results are better than previous works, but it does not convey how is this achieved, while the figure in the appendix shows both better results and how are these achieved. This is also a suggestion and it is up to the authors to make the change or not.
> > 4. Ok!
> > 5. I do not find where it is mentioned in the appendix what is the range of hyperparameters and that for all the models it was use the same gird search. Is this written anywhere in the paper?

---

> > > ### Author Response · Authors · 2022-09-27
> > > **Response to new comments from Reviewer Mvt1**
> > >
> > > We thank you for your feedback. In what follows we address each of your comments. The requested changes are highlighted in blue in the most recent version of our submission.
> > > 1. **Showing the presence of  object dissimilarity in the ground-truth saliency maps:** This requires statistically measuring or  qualitatively visualising multiple dissimilarity cues, namely, object, size, texture etc, which is difficult given the complex scenes that current deep learning techniques deal with.  However applying these current techniques on simpler scenes, as was done in classical saliency techniques [1,2], shows that dissimilarity cues are significant for saliency prediction (please refer to page 24, figure 9, row 4).
> > > Note that, while the classical saliency techniques [1] used low- level contrast cues in *simple scenes* to show that saliency is visually dissimilar, the current deep learning methods learn to predict the saliency on *complex scenes* that are dependent on multiple dissimilarity cues. Therefore, statistically measuring or qualitatively showing the relationship of *multiple* dissimilarity cues in the ground-truth saliency maps is infeasible.
> > >  We are not aware of any such statistics but we would be grateful if the reviewer kindly points us to any such statistical metric.
> > > 2. **Hyperparameters training details:** We have updated this on pages 17-18 (highlighted in blue) of the latest version of our submission. Apologies for not updating this in the text of the submission, previously.
> > > 3. **Figure 1 update:** We have updated the figure 1 as per your suggestion.
> > > Thank you, once again, for your time and we await hearing from you.
> > >
> > > **References:**
> > > 1. Radhakrishna Achanta and Sabine Susstrunk, Saliency Detection using Maximum Symmetric Surround, International Conference on Image Processing (ICIP), Hong Kong, September 2010.
> > > 2. Gokhan Yildirim and Sabine Süsstrunk (2014). FASA: Fast, Accurate, and Size-Aware Salient Object Detection. In Computer Vision - ACCV 2014.

---

> > > > ### Comment · Reviewer_Mvt1 · 2022-09-28
> > > > **Response**
> > > >
> > > > 1. Showing the presence of object dissimilarity in the ground-truth saliency maps: This could be done by finding a statistical relationship between the object dissimilarity of each object (eg. by taking Eq.(2) normalized) and the saliency of the object. This could have been explored with a scatter plot where each point would represent an object in the dataset, the x-axis would Eq.(2) and the y-axis the average saliency of the object. The scatter plot may show how dissimilarity relates with saliency. The same could be done with the size.
> > > > 2. Hyperparameters: it is not clear what batch sizes were included in the grid search. How many learning rates were tried from the reported interval?
> > > > 3. Figure 1: Ok!

---

> > > > > ### Author Response · Authors · 2022-10-07
> > > > > **Response to the latest comments from Reviewer Mvt1**
> > > > >
> > > > > We thank you, once again, for your feedback. We address each of your comments, hereunder. The requested changes have been incorporated in the most recent version of our submission.
> > > > > 1. **Showing the presence of object dissimilarity in the ground-truth saliency maps:** We have updated the paper with 2 plots, showing the presence of appearance dissimilarity and size in the ground-truth saliency maps (please refer to Figure 2). We have also referred to Figure 2 in our text (highlighted in blue) on page 2. To obtain these plots, we have used the same calculation as in Section 3.2 and have used the log scale for better visualisation.
> > > > > **Statistical clarification:**
> > > > > To validate the plots statistically, we calculate Spearman's correlation to determine the relationship between both appearance dissimilarity and size with the saliency values, respectively. There is a strong, positive monotonic correlation between size and saliency ($r_s=0.817, p < .001$). The correlation between appearance dissimilarity and saliency is weaker but still statistically significant ($r_s=0.295, p < .001$).
> > > > >
> > > > > 2. **Hyperparameters:** We have also specified the details of the batch sizes and the learning rate increment in the reported interval. Apologies, for not being clear previously!
> > > > >
> > > > > We hope that we have successfully addressed all of your comments. We are looking forward to hearing from you

---

### Review · Reviewer_RUH5 · 2022-08-11

**Summary Of Contributions:**

This paper argues for the use of object-level dissimilarity cues to improve the performance of saliency estimators. They create dissimilarity cues with the help of an object detector, replacing the boxes with rectangular masks which have values indicating some dissimilarity signals: appearance dissimilarity (inverse cosine similarity of the appearance features), and size (relative to the image). They show that adding these cues into existing architectures before the decoding step increases performance by a small (but consistent) margin on various metrics across 3 base models.


**Requested Changes:**

I would love to see the effect of (1) reordering the averages so that we use mean features when computing dissimilarity (to simplify the method) and (2) not averaging within objects when computing dissimilarity (since this finer-grained info is readily available and not yet used).

> "we slice the features in the last SSD layers using the predicted bounding box"

I think "crop" is the word usually used to describe this operation.

> "Despite this, these methods fail to account for the dissimilarity between objects, which humans naturally do."

This sentence needs work. Maybe writing it in reverse would be better (e.g, "For humans, differences between objects affect saliency, while current methods essentially deal with all objects independently".)

high-level one -> high-level ones



**Strengths And Weaknesses:**

The paper is well-written and easy to understand. The idea of using dissimilarity cues makes sense, and the implementation here is straightforward. The presentation, with helpful figures right next to where they are needed, is excellent.

The size dissimilarity metric (Eq. 3) seems like it is just a size metric (relative to the image, not relative to other objects). It is plausible that indicating size to the model allows it to tease out dissimilarity, but I think it's more likely that this simply tilts preferences toward larger objects, which is not what the text suggests.

I am not sure why Eq. 1 compares the features at corresponding positions (k) in the boxes. Why not just compute an average feature within the box, and compute one cosine similarity? We take a sum anyway, so I doubt the per-pixel step is adding much. With that said, I wonder if finer-grained dissimilarity scores could actually be useful here (instead of discarding it all, with the averages). The airplane in Figure 3 is a good example of this -- so much white area in the box is actually not salient. So, what would we get with per-pixel similarity (using "pixels"/features on the same SSD feature map)? This brings us closer to a low-level contrast cue, but maybe that's the key anyway.

---

> ### Author Response · Authors · 2022-08-29
> **Response to Reviewer RUH5**
>
> Thank you for your time to review our work and for all your valuable feedback. We have incorporated
> your different comments in our updated version, and re-worded/corrected some unclear sentences. For
> better readability of our revision, we have *highlighted in blue* the main modifications we have made.
> Below, we respond to each of your comments.
> 1. **Effect of averaging of features.** We show the performance of our method under the following
> two experimental settings in Section H, Table 11 and Figure 9 in the appendix of the revised
> paper (pages 23-24): a) Reordering the channel vs. spatial mean operation to leverage the mean
> features to compute dissimilarity, where the spatial average within the bounding box is taken
> and then the cosine dissimilarity across the features is calculated; b) No averaging within the
> detected boxes to compute the dissimilarity. We show both quantitatively and qualitatively
> that Our method, where we compute the dissimilarity score between the channel-wise averaged
> features followed by the spatial averaging within the bounding box, outperforms both setup a
> and b. Further details are provided in the appendix of our revised paper.
> 2. **Size dissimilarity.**  We calculate the size dissimilarity as a ratio of the object bounding box to
> the image size. We do not calculate the size of the masks between the objects as this would assign
> higher dissimilarity values to small objects in the presence of other larger objects from the same
> category, which differs from the way humans view salient objects in a scene [1]. Specifically, when
> there are multiple objects from the same category but with different sizes, human attention tends
> to focus on the objects with larger sizes. This is what we mimic in our model. However, when
> objects of different categories are present, the model uses both appearance and size dissimilarities
> to predict the saliency, which is again aligned with human attention. We have clarified the text
> at the end of Section 3.2, page 6, in the revised version of the main paper.
> 3. **Rephrasing and sentence clarification.** We have taken these into consideration and have
> edited the paper, with changes highlighted in blue.
>
> **References:**
> 1. Talia Konkle and Aude Oliva. A Real-World Size Organization of Object Responses in Occipitotemporal Cortex. Neuron,74(6):1114–1124, 2012.
>
> We hope that we have successfully addressed all of your comments. We are looking forward to hearing
> from you

---

> > ### Comment · Reviewer_RUH5 · 2022-09-02
> > **OK**
> >
> > This is helpful, thank you. I have a couple small concerns.
> >
> > I am a bit confused by a phrase in the added text:
> > > [our model] uses the dissimilarity score between the channel-wise averaged features before spatially averaging
> > them
> >
> > What do you mean channel-wise averaged features? I think there is no average across the channel dimension, as this would ruin the cosine similarity.
> >
> > In Figure 9d, why do we see such pronounced vertical streaks? My guess is that this is an artifact of resizing the crops to tiny squares (before similarity), and then resizing back to the original dimensions (after similarity). Can you clarify the details here? I'm thinking that using larger squares might give better results.

---

> > > ### Author Response · Authors · 2022-09-06
> > > **Response to new comments from Reviewer RUH5**
> > >
> > > Thank you for your valuable feedback. Below we respond to each of your comments.
> > >
> > > 1. Apologies, it was not clearly worded in the caption of figure 9. Yes, we do not average features channel-wise. We have rephrased this in section H, Page 23 and in the caption of Figure 9, Page 24  for better clarity.
> > >
> > > 2. We have changed the order of figures in Figure 9 for better readability. Note that Figure 9d in the previous version is Figure 9b in the latest version.
> > > Specifically, for the figure 9b (w/o averaging results), we cannot use larger squares as the bounding box resolution comes from the object detector. This is consistent across all the experimental setups (i.e. w/o averages, reordered averages and Ours). Nonetheless, Our method outperforms both the no-averaging and reordered averaging setup (Table 11, Page 23).
> > >
> > > We hope that we have successfully addressed all of your comments and we thank you again for your valuable time. We are looking forward to hearing from you.

---

> > > > ### Comment · Reviewer_RUH5 · 2022-09-07
> > > > **Thanks**
> > > >
> > > > Thanks, this addresses my concerns.

---

### Review · Reviewer_eP49 · 2022-08-15

**Summary Of Contributions:**

This paper presents a new scheme for deep saliency prediction. Saliency prediction is the task of predicting the part of an image that would get the most attention from a human viewer. The task has been addressed in a long line of work; however, the key insights that this work brings to bear is to incorporate: object and size dissimilarity as part of an end-to-end pipeline. For object dissimilarity, the idea is that an object's appearance determines how distinctive the object appears in a scene. To model object dissimilarity, the proposed scheme first passes the input image into an object detection module to obtain bounding boxes for all objects in the image. An embedding feature vector is then obtained for each object in the scene, and used in a cosine-like similarity metric. This score is propagated into a feature map, the same size as the input, where the spatial locations of the objects correspond to the previously calculated metric per-object. A dissimilarity mask is obtained for each object in the scene. Similarly, a normalized size (bounding box area divided by image area) is calculated for each object. A size mask is also created for each object in the scene  as well. All of the masks are then concatenated and fed into a saliency decoder. The end-to-end scheme is then trained using a loss that is the KL-divergence between the network's predicted saliency mask, and ground-truth saliency masks. The proposed scheme is then demonstrated on 3 datasets: SALICON, MIT1003, and CAT2000 with effective performance across these settings.

**Broader Impact Concerns:**

None.

**Requested Changes:**

I am fine to accept this paper, as is, without any major changes. However the authors can see the weaknesses section for issues they can address.

**Strengths And Weaknesses:**

### Strengths

- **Thorough experimental demonstration**: This paper thoroughly demonstrates the effectiveness of the proposed scheme on several datasets. The findings indicate that the proposed scheme outperforms or matches current state-of-the-art approaches on these datasets. In addition, the ablation experiments performed in this work are also important. The ablation studies indicate that the size and object features provide a modest performance improvement on the saliency detection task.

- **Clear Writing**: The writing in this paper is clear and easy to follow. The authors do a great job of taking a reader along.

- **Related Work**: The related work in this paper is quite thorough and comprehensive. Importantly, the key differentiator between this paper and other works is very clearly mentioned in the introduction and the rest of the paper.

### Weaknesses
First, I want to note that I donot have any disqualifying changes for this manuscript. I am fine to accept the manuscript as is. These points are mostly just observations about the paper.

- **improvement from object and size features is modest**: I expected the performance improvement due to incorporating the size and object features to be more substantial, but the effect across the datasets considered is quite modest. In some cases, it is on the order of 0.02 improvement, which is quite small. This tells me that perhaps the datasets and/or the task are somewhat saturated and maybe even solved for the settings considered. It could be that on a more challenging dataset or setting the effects of the proposed additions might be larger.

- **What is the point of the SVCCA part in the appendix?**: In the section where the authors apply SVCCA to the representations, it is unclear what the goal is there? Perhaps a clarification would be helpful. However, this section is not critical to the work. In addition, SVCCA has some known limitations, so CKA (Similarity of neural network representations revisited) might be a better alternative here.

---

> ### Author Response · Authors · 2022-08-29
> **Response to Reviewer eP49**
>
> Thank you for your time to review our work and for all your valuable feedback. Below are our responses
> to your comments.
> 1. **Saturated datasets**. The datasets used in this work are consistent with all the baseline papers
> and are the standard benchmarks for the task of saliency prediction. To account for more chal-
> lenging datasets, we show results on CAT2000 in Table 2, page 10 of the paper.
>  **Modest improvement on some metrics:** On the SALICON benchmark, the modest im-
> provements in some metrics, particularly AUC, are due to this metric being saturated and not
> giving a detailed measure of the predictions’ quality, as discussed in [1]. Therefore, as mentioned
> in Section 4.1.2, page 9 of the paper, we evaluate our method and the baselines using the sAUC
> metric, which overcomes the saturation issue of the AUC. Note that we report the results on six
> different saliency metrics as each metric has a different range and deviation [2].
> 2. **Why SVCCA?** We report the performance of our model versus the baselines using SVCCA,
> shown in Table 7, page 19 (appendix), to show the effect of different dissimilarity metrics on
> appearance dissimilarity. Our results show that our model generalizes well across different dis-
> similarity metrics, ultimately obtaining comparable performances for all of them. Note that
> SVCCA is scale-invariant and captures the semantic proximity of different classes, with similar
> classes having similar sensitivities. We would like to thank the reviewer for their suggestion of
> using CKA. We will consider it as a possible future direction.
>
> **References**:
> 1. Zoya Bylinskii, Adrià Recasens, Ali Borji, Aude Oliva, Antonio Torralba, Frédo Durand. Where
> Should Saliency Models Look Next? Computer Vision– ECCV 2016.
> 2. Matthias Kümmerer, Thomas S. A. Wallis, Matthias Bethge. Saliency Benchmarking Made
> Easy: Separating Models, Maps and Metrics. Computer Vision– ECCV 2018.
>
> We hope that we have successfully addressed all of your comments. We are looking forward to hearing
> from you

---

> > ### Comment · Reviewer_eP49 · 2022-09-14
> > **Response addresses my concerns**
> >
> > Thanks for the updates, the response addresses the concerns I had. Since the SVCCA portion is not key to the work, I am fine with the response on that front.

---

### Decision · Action_Editors · 2022-11-03

**Recommendation:** Accept as is

**Comment:**

All three reviewers were broadly in favor of the paper even in its first iteration. Each reviewer had an individual set of concerns: RUH5 had questions about a few of the technical details that was clarified cleanly in a the revision; eP49 was in favor of acceptance as is, but had some observations that were addressed in discussion; Mvt1 had more substantial questions about experiments that were resolved by a discussion with multiple back-and-forths and a revision.

Post-revisions, two out of three reviewers recommended a clear accept and the third leaned accept. Having examined the manuscript, the discussion, and the individual reviews, the AE is in favor of acceptance as is (other than, of course, removing the blue text in the final revision). The authors, in the AE's view have fully addressed all of the concerns of the reviewers. The AE would like to thank both the authors and the reviewers for engaging in thoughtful and productive discussions throughout the process.



**Audience:**

Yes. There is a strong community that is interested in understanding saliency and a broader community that uses saliency as an input for various tasks  for handling images. The proposed approach is of clear interest to both these communities. Given that the proposed system works on multiple base systems, this paper will be of clear interest. Moreover, there are people in the neuroscience community who may be interested in this paper as well.

**Claims And Evidence:**

Yes. The submission's claims are supported by clear experimental evidence that all three reviewers have pointed to and appreciated. The method is demonstrated with multiple base networks and multiple datasets.